# Online GNN Evaluation Under Test-time Graph Distribution Shifts

**Xin Zheng**
Monash University
Melbourne, Australia
`xin.zheng@monash.edu`

**Dongjin Song**
University of Connecticut
Storrs, USA
`dongjin.song@uconn.edu`

**Qingsong Wen**
Squirrel AI
Bellevue, USA
`qingsongedu@gmail.com`

**Bo Du**
Wuhan University
Wuhan, China
`dubo@whu.edu.cn`

**Shirui Pan**[*]
Griffith University
Queensland, Australia
`s.pan@griffith.edu.au`

## Abstract

Evaluating the performance of well-trained GNN models on real-world graphs is a pivotal step for reliable GNN online deployment and serving. Due to a lack of test node labels and unknown potential training-test graph data distribution shifts, conventional model evaluation encounters limitations in calculating performance metrics (*e.g.*, test error) and measuring graph data-level discrepancies, particularly when the training graph used for developing GNNs remains unobserved during test time. In this paper, we study a new research problem, *online GNN evaluation*, which aims to provide valuable insights into the well-trained GNNs' ability to effectively generalize to real-world unlabeled graphs under the test-time graph distribution shifts. Concretely, we develop an effective **LE**arning **BE**havior **D**iscrepancy score, dubbed **LeBeD**, to estimate the test-time generalization errors of well-trained GNN models. Through a novel GNN re-training strategy with a parameter-free optimality criterion, the proposed LeBeD comprehensively integrates learning behavior discrepancies from both node prediction and structure reconstruction perspectives. This enables the effective evaluation of the well-trained GNNs' ability to capture test node semantics and structural representations, making it an expressive metric for estimating the generalization error in online GNN evaluation. Extensive experiments on real-world test graphs under diverse graph distribution shifts could verify the effectiveness of the proposed method, revealing its strong correlation with ground-truth test errors on various well-trained GNN models.[1]

## 1 Introduction

Recent advances in graph neural networks (GNNs) have achieved great success with promising learning abilities for various graph structural data related applications in the real world (Zhang et al., 2022; Jin et al., 2023a; Zheng et al., 2022a;b; Jin et al., 2022; Liu et al., 2023c; Zheng et al., 2022c; 2023d; Luo et al., 2023a; 2024; Zhang et al., 2023a). The ultimate goal in developing GNN models is to achieve practical deployment and serving, with the expectation that well-trained GNNs will show expressive generalization ability when applied to diverse real-world, unseen, and unlabeled graph data Zheng et al. (2023a;c;b); Wu et al. (2024a); Zhang et al. (2023b;c); Liu et al. (2023a); Tan et al. (2023). In this case, understanding and evaluating the performance of well-trained GNN models becomes an essential and pivotal step for reliable GNN deployment in practice. For example, in real-world financial transaction networks (Amazon, 2021), model designers strive to enhance the capacity of their well-trained GNNs to identify newly emerging suspicious transactions. Meanwhile,

---

[*]Corresponding Author

[1]Code is available at `https://github.com/Amanda-Zheng/LEBED`

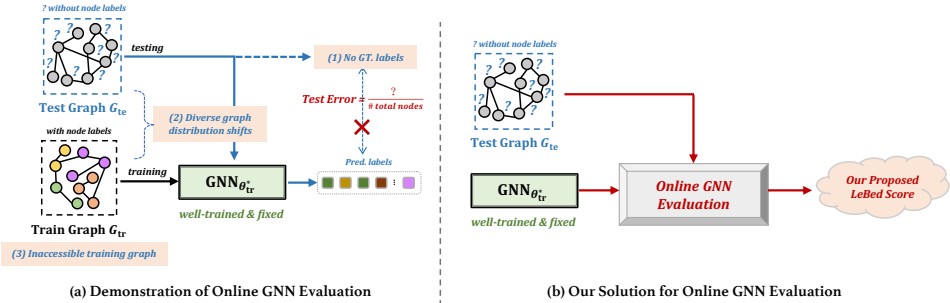

(a) Demonstration of Online GNN Evaluation    (b) Our Solution for Online GNN Evaluation

Figure 1: Illustration of the proposed online GNN evaluation problem and our solution.

users seek confidence in these well-trained GNNs to detect and flag potentially dubious transactions within their own financial networks.

Conventionally, model evaluation involves using a thoroughly annotated test graph with labels to assess the model's performance, and this assessment is accomplished by comparing the model's predicted labels to the provided ground-truth labels, allowing the calculation of accuracy (or test error) as a pivotal metric. However, this strategy falls short in practical GNN model evaluation scenarios. Typically, test graph data lacks ground-truth node labels since annotating it could be prohibitively expensive and inefficient, making it infeasible to compute the accuracy metric. Moreover, considering the natural non-independent and non-identically distributed characteristic of graph data, there could be various unknown distribution shifts between the training graphs and real-world test graphs Wu et al. (2024b). For instance, the practical test graph data might be collected with different procedures, domains, and time Liu et al. (2023b); Luo et al. (2023b); Pan et al. (2024), leading to distinct node contexts, graph structures, and scales, with diverse *node feature shifts* (Jin et al., 2023b), *domain shifts (Wu et al., 2020)*, and *temporal shifts (Wu et al., 2022)*. In light of this, one possible solution to model evaluation involves quantifying the training-test discrepancy in graph data level. This can be accomplished by employing distribution measurement functions, *e.g.*, maximum mean discrepancy (MMD) (Deng & Zheng, 2021), to calculate representation-level differences between the training and test graphs, serving as essential features for evaluating the GNN's performance. However, in more practical *online* GNN serving scenarios, the training graph utilized for developing the GNN models is often inaccessible due to privacy constraints, making it unfeasible to directly estimate the graph data-level disparity.

Given all these real-world circumstances involving: (1) unlabeled and unseen test graph data; (2) potential training-test graph distribution shifts; and (3) inaccessible training graph data for online scenarios, an intriguing problem emerges: *Is it possible to estimate the performance of a well-trained GNN model on test graph data affected by distribution shifts, without access to its ground truth labels and original training graph data?*

In this work, we first provide a feasible solution to answer this question by identifying it as an **online GNN evaluation** problem as shown in Fig. 1. Concretely, given a well-trained GNN model, online GNN evaluation aims to accurately estimate the well-trained GNN's generalization error on real-world unlabeled test graphs under distribution shifts, without needing to access the training graph data. To this end, we propose to estimate the **LE**arning **BE**havior **D**iscrepancy score between the training and test graphs from the view of GNN model training, dubbed as **LEBED**. More specifically, we develop a novel GNN re-training strategy with a parameter-free optimality criterion, to re-train a new GNN on the test graph for effectively modeling the training-test learning behavior discrepancy. With the guidance of both node prediction discrepancy and structure reconstruction discrepancy, the proposed LEBED score computes the distance between the optimal weight parameters of the test graph re-trained GNN *vs.* the training graph well-trained GNN, to evaluate the well-trained GNN's ability to capture node semantics and structural representations at test time. Consequently, the proposed LEBED score can serve as an expressive generalization error metric of the well-trained GNN for online GNN evaluation. It is important to note that the core of this work is to estimate well-trained GNN models' performance, rather than developing new GNN models to improve the generalization ability. Throughout the entire process of online GNN evaluation, the well-trained GNN models would remain fixed. In summary, the contributions of this work are as follows:

• **Problem.** We study a new research problem, online GNN evaluation, which aims to provide valuable insights into well-trained GNNs' ability to effectively generalize to real-world, unlabeled test graphs under test-time distribution shifts.

- **Solution.** We develop an effective learning behavior discrepancy score, dubbed as **LeBed**, to estimate the test-time generalization error of a well-trained GNN model for online GNN evaluation. Through a novel GNN re-training strategy with a parameter-free optimality criterion, LeBed comprehensively integrates training-test learning behavior discrepancies from both node prediction and structure reconstruction perspectives, enabling it an expressive metric for effective online GNN evaluation.
- **Evaluation.** We evaluate the proposed method on real-world unseen, unlabeled test graphs under diverse graph distribution shifts. Extensive experimental results reveal strong correlations with ground-truth test errors on various well-trained GNN models, providing compelling evidence for the efficacy of the proposed method.

**Prior Works**. Our research is related to existing studies on *predicting model generalization error* (Deng & Zheng, 2021; Garg et al., 2022; Yu et al., 2022; Guillory et al., 2021; Deng et al., 2021; Yu et al., 2022), which aims to estimate a model's performance on unlabeled data from the unknown and shifted distributions. However, these researches are designed for data in Euclidean space (*e.g.*, images) while our research is specifically designed for graph structural data, with a particular focus on applications in online deployment scenarios. Our research also significantly differs from others in unsupervised graph domain adaption (Yang et al., 2021; Zhang et al., 2019; Wu et al., 2020), out-of-distribution (OOD) generalization (Li et al., 2022; Zhu et al., 2021). More detailed related work can be found in Appendix A.

## 2 THE PROPOSED METHOD

**Preliminary.** At the stage of GNN development, we have a fully-observed training graph $G_{\mathrm{tr}} = (\mathbf{X}_{\mathrm{tr}}, \mathbf{A}_{\mathrm{tr}}, \mathbf{Y}_{\mathrm{tr}})$ with the number of $N$ nodes with $C$-classes of node labels, where $\mathbf{X}_{\mathrm{tr}} \in \mathbb{R}^{N \times d}$ is the $d$-dimension nodes feature matrix indicating node attribute semantics, $\mathbf{A}_{\mathrm{tr}} \in \mathbb{R}^{N \times N}$ is the adjacency matrix indicating whether nodes are connected or not by edges with $\mathbf{A}_{\mathrm{tr}}^{i,j} = \{0, 1\} \in \mathbb{R}$ for $i$-th and $j$-th nodes, and $\mathbf{Y}_{\mathrm{tr}} \in \mathbb{R}^{N \times C}$ denotes the node labels.

• *Training Stage.* A GNN model is trained on $G_{\mathrm{tr}}$ according to the following objective function for node classification:

$$
\begin{aligned}
\boldsymbol{\theta}_{\mathrm{tr}}^* &= \min_{\boldsymbol{\theta}_{\mathrm{tr}}} \mathcal{L}_{\mathrm{cls}}\left(\hat{\mathbf{Y}}_{\mathrm{tr}}, \mathbf{Y}_{\mathrm{tr}}\right), \text{ where} \\
\mathbf{Z}_{\mathrm{tr}}, \hat{\mathbf{Y}}_{\mathrm{tr}} &= \mathrm{GNN}_{\boldsymbol{\theta}_{\mathrm{tr}}}(\mathbf{X}_{\mathrm{tr}}, \mathbf{A}_{\mathrm{tr}}).
\end{aligned}
\tag{1}
$$

The parameters of GNN trained on $G_{\mathrm{tr}}$ is denoted by $\boldsymbol{\theta}_{\mathrm{tr}}$, $\mathbf{Z}_{\mathrm{tr}} \in \mathbb{R}^{N \times d_1}$ is the output node embedding of graph $G_{\mathrm{tr}}$ from $\mathrm{GNN}_{\boldsymbol{\theta}_{\mathrm{tr}}}$, and $\hat{\mathbf{Y}}_{\mathrm{tr}} \in \mathbb{R}^{N \times C}$ denotes the output node labels predicted by the trained $\mathrm{GNN}_{\boldsymbol{\theta}_{\mathrm{tr}}}$. By optimizing the node classification loss function $\mathcal{L}_{\mathrm{cls}}$ (*e.g.*, cross-entropy loss) between GNN predictions $\hat{\mathbf{Y}}_{\mathrm{tr}}$ and ground-truth node labels $\mathbf{Y}_{\mathrm{tr}}$, the GNN model that is well-trained on $G_{\mathrm{tr}}$ can be denoted as $\mathrm{GNN}_{\boldsymbol{\theta}_{\mathrm{tr}}^*}$ with optimal weight parameters $\boldsymbol{\theta}_{\mathrm{tr}}^*$. Note that once we obtain the optimal $\mathrm{GNN}_{\boldsymbol{\theta}_{\mathrm{tr}}^*}$ that has been well-trained on $G_{\mathrm{tr}}$, the GNN model would be **fixed** and $G_{\mathrm{tr}}$ would not be accessible during test time for online evaluation.

• *Test Time.* For online GNN deployment and serving, given a real-world unlabeled test graph $G_{\mathrm{te}} = (\mathbf{X}_{\mathrm{te}}, \mathbf{A}_{\mathrm{te}})$ including $M$ nodes with its feature matrix $\mathbf{X}_{\mathrm{te}} \in \mathbb{R}^{M \times d}$ and its adjacency matrix $\mathbf{A}_{\mathrm{te}} \in \mathbb{R}^{M \times M}$, we assume that there are potential distribution shifts between $G_{\mathrm{tr}}$ and $G_{\mathrm{te}}$, which mainly lies in node contexts, graph structures, and scales, but the label space keeps consistent under the covariate shift, *i.e.*, all nodes in $G_{\mathrm{te}}$ are constrained in the same $C$-classes as $G_{\mathrm{tr}}$. Generally, the test error of node classification produced by the well-trained $\mathrm{GNN}_{\boldsymbol{\theta}_{\mathrm{tr}}^*}$ can be calculated as:

$$
\text{Test Error}(G_{\mathrm{te}}, \mathrm{GNN}_{\boldsymbol{\theta}_{\mathrm{tr}}^*}) = \frac{1}{M} \sum_{i=1}^{M} \mathbf{1}\left\{\hat{y}_{\mathrm{te}}^i \neq y_{\mathrm{te}}^i\right\},
\tag{2}
$$

which indicates the percentage of incorrectly predicted node labels between the GNN predicted labels $\hat{y}_{\mathrm{te}}^i \in \hat{\mathbf{Y}}_{\mathrm{te}}$ and *'ground truths'* $y_{\mathrm{te}}^i \in \mathbf{Y}_{\mathrm{te}}$, where $\mathbf{1}\{\cdot\}$ is the indicator function. However, during the practical test time, the true node labels $\mathbf{Y}_{\mathrm{te}}$ for unseen test graphs are typically **unavailable**, making the computation of the test error infeasible. Consequently, there is an urgent need to devise a metric or a score that can serve as a substitute for the test error, to assess the performance of a well-trained GNN on unseen and unlabeled test graphs for real-world online GNN deployment.

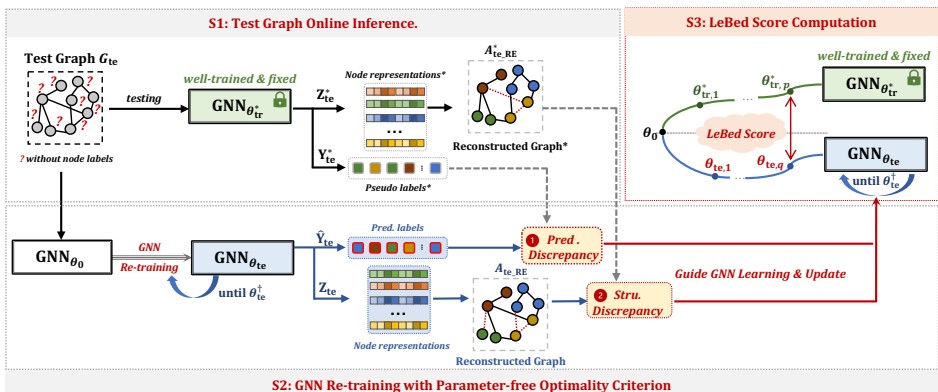

Figure 2: The overview of the proposed LEBED score for online GNN evaluation under test-time distribution shifts, including three steps, *i.e.*, S1: Test graph online inference; S2: GNN re-training with parameter-free optimality criterion; S3: LEBED score computation. All ∗ superscripts indicate the corresponding variables would remain fixed for online GNN evaluation.

## 2.1 PROBLEM FORMULATION

At the test time, given the model $\text{GNN}_{\boldsymbol{\theta}_{\text{tr}}^*}$ that has been well-trained on $G_{\text{tr}}$, and a real-world test graph $G_{\text{te}} = (\mathbf{X}_{\text{te}}, \mathbf{A}_{\text{te}})$ without node labels as the input, the **goal** of **online GNN evaluation** is to learn a test error estimation model $f_{\boldsymbol{\phi}}(\cdot)$ parameterized by $\boldsymbol{\phi}$ for obtaining a score as:

$$\text{Score} = f_{\boldsymbol{\phi}}(G_{\text{te}}, \text{GNN}_{\boldsymbol{\theta}_{\text{tr}}^*}) \propto \text{Test Error}, \tag{3}$$

where $f_{\boldsymbol{\phi}}(G_{\text{te}}, \text{GNN}_{\boldsymbol{\theta}_{\text{tr}}^*}) \to a$ and $a \in \mathbb{R}$ is a scalar score indicating the test error of the well-trained GNN on $G_{\text{te}}$. More specifically, the score should exhibit a direct correlation relationship $\propto$ with the test error according to Eq. (2).

It is important to note that several critical elements govern the online GNN evaluation: (1) the test graph $G_{\text{te}}$ remains unobserved during the training phase and it lacks node label annotations; (2) there are potential unknown graph data distribution shifts between $G_{\text{tr}}$ and $G_{\text{te}}$ that could challenge GNNs' inference ability; (3) the training graph $G_{\text{tr}}$ used for training $\text{GNN}_{\boldsymbol{\theta}_{\text{tr}}^*}$ cannot be observed due to privacy constraints in online scenarios.

## 2.2 LEBED: LEARNING BEHAVIOR DISCREPANCY SCORE

To achieve the goal of online GNN evaluation, in this work, we first comprehensively exploit three-fold informative aspects from both the well-trained GNN and the test graph: (1) *GNN model architecture*, with its well-trained model weight parameters; (2) *output node representations*, which reflect the well-trained GNN's capacity to capture the test graph node semantics and graph structures; (3) *output node pseudo labels*, which indicates the ability of the well-trained GNN to project the captured test graph contexts to the node class label space. By thoroughly utilizing such information, in this work, we develop a learning behavior discrepancy score, *i.e.*, LEBED score, to estimate the test-time generalization error of a well-trained GNN model on real-world unseen, unlabeled, and potentially distribution-shifted test graphs.

Concretely, we propose a GNN re-training strategy with a parameter-free optimality criterion to associate the graph data-level discrepancy, with the GNN model-level learning behavior discrepancy between the training and test graphs. To model the GNN learning behavior discrepancy comprehensively, we incorporate two-fold distinct discrepancies: node prediction discrepancy, denoted as $D_{\text{Pred.}}$, and structure reconstruction discrepancy, denoted as $D_{\text{Stru.}}$. More specifically, $D_{\text{Pred.}}$ quantifies the disparity between the pseudo-labels generated by GNN models trained on the training graph and those re-trained on the practical test graph. Meanwhile, $D_{\text{Stru.}}$ evaluates the discrepancy in the quality of output node representations by observing their ability to reconstruct the structure of the test graph. Subsequently, we utilize $D_{\text{Pred.}}$ as the supervision signal for instructing the GNN re-training process on the test graph, while $D_{\text{Stru.}}$ serves as the self-supervised iteration stop criterion, signaling the potential optimal solution of the re-trained GNN. At last, we derive the LEBED score based on the distance between the optimal weight parameters of the test graph re-trained GNN *vs.*

training graph well-trained GNN, so that the proposed LEBED score can serve as an effective metric of the generalization error for the well-trained GNN on the test graph. The overall framework of computing the proposed LEBED score for online GNN evaluation is illustrated in Fig 2. The detailed procedure of the proposed method is outlined as follows.

**S1: Test Graph Online Inference.** Given a well-trained GNN model with known initialization $\boldsymbol{\theta}_0$, model architectures $\mathrm{GNN}_{\boldsymbol{\theta}}$, and optimal model parameters $\boldsymbol{\theta}_{\mathrm{tr}}^*$ obtained on the training graph, for the first step, we take the real-world unlabeled test graph $G_{\mathrm{te}} = (\mathbf{X}_{\mathrm{te}}, \mathbf{A}_{\mathrm{te}})$ as the input of the online-deployed model $\mathrm{GNN}_{\boldsymbol{\theta}_{\mathrm{tr}}^*}$, so that we could obtain the output node representations $\mathbf{Z}_{\mathrm{te}}^* \in \mathbb{R}^{M \times d_1}$ and node class pseudo labels $\mathbf{Y}_{\mathrm{te}}^* \in \mathbb{R}^{M \times C}$ as

$$\mathbf{Z}_{\mathrm{te}}^*, \mathbf{Y}_{\mathrm{te}}^* = \mathrm{GNN}_{\boldsymbol{\theta}_{\mathrm{tr}}^*}(\mathbf{X}_{\mathrm{te}}, \mathbf{A}_{\mathrm{te}}). \tag{4}$$

In this case, the following primary focus, for assessing the performance of $\mathrm{GNN}_{\boldsymbol{\theta}_{\mathrm{tr}}^*}$ on the real-world test graph, should revolve around evaluating two critical aspects, *i.e.*, the capacity of the output node representations $\mathbf{Z}_{\mathrm{te}}^*$ containing well-captured node semantics and structure information of the test graph, and the quality of output node pseudo labels $\mathbf{Y}_{\mathrm{te}}^*$.

**S2: GNN Re-training With Parameter-free Optimality Criterion.** Given the obtained pseudo labels $\mathbf{Y}_{\mathrm{te}}^*$ and node representations $\mathbf{Z}_{\mathrm{te}}^*$, for the second step, we initialize a new GNN model $\mathrm{GNN}_{\boldsymbol{\theta}_{0,\mathrm{te}}}$ with the same model architectures as the well-trained GNN model, *i.e.*, $\boldsymbol{\theta}_{0,\mathrm{te}} = \boldsymbol{\theta}_{0,\mathrm{tr}}$. We use $\boldsymbol{\theta}_0$ for simplification in the following. Then, we re-train the new $\mathrm{GNN}_{\boldsymbol{\theta}_0}$ from scratch to fit the unseen test graph data. Importantly, the supervision signal comes from the pseudo labels $\mathbf{Y}_{\mathrm{te}}^*$ to quantifies the node prediction discrepancy, *i.e.*, $D_{\mathrm{Pred.}} = \mathcal{L}_{\mathrm{cls}}\left(\hat{\mathbf{Y}}_{\mathrm{te}}, \mathbf{Y}_{\mathrm{te}}^*\right)$, as

$$\boldsymbol{\theta}_{\mathrm{te}}^\dagger = \min_{\boldsymbol{\theta}_{\mathrm{te}}} \mathcal{L}_{\mathrm{cls}}\left(\hat{\mathbf{Y}}_{\mathrm{te}}, \mathbf{Y}_{\mathrm{te}}^*\right), \text{ where}$$
$$\mathbf{Z}_{\mathrm{te}}, \hat{\mathbf{Y}}_{\mathrm{te}} = \mathrm{GNN}_{\boldsymbol{\theta}_{\mathrm{te}}}(\mathbf{X}_{\mathrm{te}}, \mathbf{A}_{\mathrm{te}}). \tag{5}$$

By minimizing the node classification loss in the re-training with $D_{\mathrm{Pred.}}$, we could iteratively optimize the re-trained model from $\mathrm{GNN}_{\boldsymbol{\theta}_0}$ to *optimal* $\mathrm{GNN}_{\boldsymbol{\theta}_{\mathrm{te}}^\dagger}$ after $q$-step updates as $\{\boldsymbol{\theta}_0, \boldsymbol{\theta}_{\mathrm{te},1}, \cdots, \boldsymbol{\theta}_{\mathrm{te},q}\}$ via stochastic gradient descent. However, due to lacking the ground-truth test node class annotations in practice, it becomes challenging to ascertain whether specific iterations of GNN re-training in Eq. (5) can lead to optimal weight parameters on the test graph. In other words, it is difficult to determine the extent to which $\boldsymbol{\theta}_{\mathrm{te},q} \approx \boldsymbol{\theta}_{\mathrm{te}}^\dagger$.

In light of this, we derive a parameter-free optimality criterion that leverages the inherent structural property of graph data to facilitate the learning and updating of the re-trained GNN model. By reconstructing the test graph structure through the output node representations $\mathbf{Z}_{\mathrm{te}} = \left\{\mathbf{z}_{\mathrm{te}}^i \in \mathbb{R}^{d_1}\right\}_{i=1}^M$ at each learning step, we could obtain the explicit and accurate self-supervision signals for guiding the optimization of $\mathrm{GNN}_{\boldsymbol{\theta}_{\mathrm{te}}}$. More specifically, the proposed parameter-free criterion computes the discrepancy in the ability of reconstructing graph structures, *i.e.*, $D_{\mathrm{Stru.}}$, between the reference node representations from the well-trained GNN and the updated node representations from the re-trained GNN in each step. This is expressed as:

$$D_{\mathrm{Stru.}} = |g(s(\mathbf{Z}_{\mathrm{te}}), \mathbf{A}_{\mathrm{te}}) - g(s(\mathbf{Z}_{\mathrm{te}}^*), \mathbf{A}_{\mathrm{te}})| <= \epsilon, \tag{6}$$

where $g(\cdot)$ measures the ability of reconstructing graph structures from the node representations, with the hyper-parameter $\epsilon$ controlling the degree of reconstruction discrepancy, and $s(\cdot)$ is the similarity-based reconstruction function for node representations. Here we use the cosine similarity function calculated as $s(\mathbf{Z}_{\mathrm{te}}) = \mathbf{Z}_{\mathrm{te}} \cdot \mathbf{Z}_{\mathrm{te}}^T / \|\mathbf{Z}_{\mathrm{te}}\|_2 \cdot \|\mathbf{Z}_{\mathrm{te}}^T\|_2 \in \mathbb{R}^{M \times M}$. Taking the first item $g(s(\mathbf{Z}_{\mathrm{te}}), \mathbf{A}_{\mathrm{te}})$ as the example, we have:

$$g(s(\mathbf{Z}_{\mathrm{te}}), \mathbf{A}_{\mathrm{te}}) = \frac{1}{M} \sum_{i=1}^M \left(\mathbf{A}_{\mathrm{te}}^{i\cdot} \cdot \log\left(\sigma\left(s(\mathbf{Z}_{\mathrm{te}})^{i\cdot}\right)\right) + \left(1 - \mathbf{A}_{\mathrm{te}}^{i\cdot}\right) \cdot \log\left(1 - \sigma\left(s(\mathbf{Z}_{\mathrm{te}})^{i\cdot}\right)\right)\right). \tag{7}$$

This calculates the binary cross-entropy objective to measure the proximity of $s(\mathbf{Z}_{\mathrm{te}})$ to the discrete graph structures $\mathbf{A}_{\mathrm{te}}$, where $i\cdot$ denotes the $i$-th row of the matrix.

Note that the rationale behind employing the proposed parameter-free criterion is based on the following considerations: firstly, the parameter-free approach eliminates the need to modify the loss

function, *i.e.*, $\mathcal{L}_{\text{cls}}$ in Eq. (5), and does not impact the re-training process. This ensures a consistent learning objective for both the re-trained $\text{GNN}_{\boldsymbol{\theta}_{\text{te}}}$ and $\text{GNN}_{\boldsymbol{\theta}_{\text{tr}}}$ for fairly measuring the proposed learning behavior discrepancy; Secondly, the parameter-free pattern avoids introducing new parameterized modules that need optimization, thereby simplifying the learning process.

**S3: LEBED Score Computation.** Given the node prediction discrepancy $D_{\text{Pred.}}$ as the optimization objective in Eq. (5), and the structure reconstruction discrepancy $D_{\text{Stru.}}$ based iteration stop criterion to indicate the optimal $\text{GNN}_{\boldsymbol{\theta}_{\text{te}}^{\dagger}}$, the proposed learning behavior discrepancy score LEBED can be calculated as:

$$\text{LEBED}\left(\text{GNN}_{\boldsymbol{\theta}_{\text{te}}^{\dagger}}(G_{\text{te}}), \text{GNN}_{\boldsymbol{\theta}_{\text{tr}}^{*}}\right) = \left\|\boldsymbol{\theta}_{\text{tr}}^{*} - \boldsymbol{\theta}_{\text{te}}^{\dagger}\right\|_{2}, \tag{8}$$

where $\|\cdot\|_{2}$ denotes the L2-norm for measuring the distance of optimal GNN model weights between that trained by the training graph and the test graph, respectively, so that it measures GNN model-level discrepancy from the view of GNN training to effectively reflect the test error during test-time online evaluation. The overall algorithm of the proposed LEBED for online GNN evaluation under test-time graph data distribution shifts can be seen in Algo. 1 in the Appendix D.

# 3 EXPERIMENTS

In this section, we verify the effectiveness of the proposed LEBED score in terms of its correlation to the ground-truth test error for online GNN evaluation on various real-world graph datasets. Concretely, we aim to answer the following questions to demonstrate the effectiveness of the proposed LEBED: **Q1:** How does the proposed LEBED perform in evaluating well-trained GNNs on online node classification under various graph distribution shifts? **Q2:** How does the proposed LEBED perform when conducting an ablation study regarding the parameter-free structure reconstruction discrepancy criterion $D_{\text{Stru.}}$? **Q3:** How sensitive are the hyper-parameter $\epsilon$ for the proposed GNN re-training strategy? **Q4:** How is the correlation between our proposed LEBED and the ground-truth test error depicted? More experimental results, discussions, and implementation details of the proposed method are provided in Appendix D.

## 3.1 EXPERIMENTAL SETTINGS

**Datasets.** We perform experiments on six real-world graph datasets with diverse graph data distribution shifts containing: node feature shifts (Wu et al., 2022; Jin et al., 2023b)), domain shifts (Wu et al., 2020), temporal shifts (Wu et al., 2022). Detailed statistics of all these datasets are listed in Table A1 in Appendix B. Note that the proposed LEBED is derived to model the statistical correlations with the ground-truth test error, so that it can be taken as the effective metric on practical unlabeled test graphs for our proposed online GNN evaluation problem. In this case, amount of test graphs with great diversity and complexity are required to observe such correlations. Hence, we propose to first adapt the raw distribution shifted graph datasets to the online GNN evaluation scenarios by extending the test graph sets in Table A1 with a blanket of 'Adapted'. More details can be found in Appendix B. For all training graphs and validation graphs, we follow the process procedures and splits in works (Wu et al., 2022) and (Wu et al., 2020).

**Online GNN Evaluation Protocol.** We evaluate four commonly used GNN models for practical online GNN deployment and serving, including GCN (Kipf & Welling, 2017), GraphSAGE (Hamilton et al., 2017) (*abbr.* SAGE), GAT (Veličković et al., 2017), and GIN (Xu et al., 2019), as well as the baseline MLP model that is prevalent for graph learning. For each model, we save its initialization and train it on the observed training graph, until the model achieves the optimal node classification on its validation set following the standard training process, so that we can obtain the 'well-trained' GNN model that keeps fixed in the online GNN evaluation process. More details of these well-trained GNN models, including architectures, training hyper-parameters, and ground-truth test error distributions, are provided in Appendix D. We report the correlation between the proposed LEBED and the ground-truth test errors under unseen and unlabeled test graphs with distribution shifts, using $R^2$ and rank correlation Spearman's $\rho$, where $R^2$ ranges $[0, 1]$, representing the degree of linear fit between two variables. The closer it is to 1, the higher the linear correlation. Spearman's $\rho$ ranges $[-1, 1]$, representing the monotonic correlation between two variables with 1 indicating the positive correlation and $-1$ indicating the negative correlation.

Table 1: Evaluation performance of well-trained GNNs on Cora and Amazon-Photo under **node feature shifts** with Spearman rank correlation $\rho$ and linear fitting $R^2$. The best results are in bold.

| Cora | GCN | | GAT | | SAGE | | GIN | | MLP | | *avg.* GNNs | |
|---|---|---|---|---|---|---|---|---|---|---|---|---|
| | $\rho$ | $R^2$ | $\rho$ | $R^2$ | $\rho$ | $R^2$ | $\rho$ | $R^2$ | $\rho$ | $R^2$ | $\rho$ | $R^2$ |
| ConfScore | 0.46 | 0.051 | 0.56 | 0.097 | 0.37 | 0.055 | 0.67 | 0.210 | 0.53 | 0.095 | 0.52 | 0.102 |
| Entropy | 0.50 | 0.070 | 0.62 | 0.121 | 0.51 | 0.094 | 0.69 | 0.234 | 0.63 | 0.154 | 0.59 | 0.134 |
| ATC-MC | -0.45 | 0.034 | -0.53 | 0.062 | -0.36 | 0.035 | -0.65 | 0.180 | -0.56 | 0.054 | -0.51 | 0.073 |
| ATC-NE | -0.51 | 0.053 | -0.61 | 0.087 | -0.53 | 0.047 | -0.69 | 0.223 | -0.67 | 0.065 | -0.60 | 0.095 |
| Thres. ($\tau = 0.7$) | -0.44 | 0.046 | -0.50 | 0.081 | -0.29 | 0.035 | -0.65 | 0.192 | -0.45 | 0.057 | -0.47 | 0.082 |
| Thres. ($\tau = 0.8$) | -0.44 | 0.057 | -0.52 | 0.101 | -0.33 | 0.048 | -0.67 | 0.227 | -0.47 | 0.071 | -0.49 | 0.101 |
| Thres. ($\tau = 0.9$) | -0.45 | 0.085 | -0.56 | 0.143 | -0.40 | 0.088 | -0.68 | 0.272 | -0.52 | 0.115 | -0.52 | 0.141 |
| **LEBED (Ours)** | **0.82** | **0.640** | **0.87** | **0.745** | **0.69** | **0.519** | **0.66** | **0.410** | **0.74** | **0.663** | **0.76** | **0.595** |

| Amazon-Photo | GCN | | GAT | | SAGE | | GIN | | MLP | | *avg.* GNNs | |
|---|---|---|---|---|---|---|---|---|---|---|---|---|
| | $\rho$ | $R^2$ | $\rho$ | $R^2$ | $\rho$ | $R^2$ | $\rho$ | $R^2$ | $\rho$ | $R^2$ | $\rho$ | $R^2$ |
| ConfScore | -0.55 | 0.251 | -0.45 | 0.173 | -0.70 | 0.376 | **0.94** | 0.870 | -0.01 | 0.012 | -0.15 | 0.336 |
| Entropy | -0.52 | 0.305 | -0.48 | 0.193 | -0.69 | 0.365 | 0.94 | 0.867 | 0.02 | 0.012 | -0.15 | 0.348 |
| ATC-MC | 0.52 | 0.220 | 0.50 | 0.180 | 0.67 | 0.326 | -0.94 | 0.850 | -0.33 | 0.015 | 0.08 | 0.319 |
| ATC-NE | 0.45 | 0.284 | 0.56 | 0.210 | 0.73 | 0.302 | -0.92 | 0.800 | -0.43 | 0.155 | 0.08 | 0.350 |
| Thres. ($\tau = 0.7$) | 0.54 | 0.223 | 0.48 | 0.174 | 0.71 | 0.376 | -0.94 | 0.859 | 0.01 | 0.008 | 0.16 | 0.328 |
| Thres. ($\tau = 0.8$) | 0.56 | 0.232 | 0.45 | 0.162 | 0.71 | 0.382 | -0.94 | 0.876 | 0.03 | 0.014 | 0.16 | 0.333 |
| Thres. ($\tau = 0.9$) | 0.57 | 0.249 | 0.42 | 0.156 | 0.71 | 0.378 | -0.94 | **0.885** | 0.05 | 0.022 | 0.16 | 0.338 |
| **LEBED (Ours)** | **0.90** | **0.730** | **0.78** | **0.616** | **0.79** | **0.425** | 0.83 | 0.705 | **0.62** | **0.530** | **0.78** | **0.601** |

**Baseline Methods.** Since our proposed LEBED is the pioneering approach for online GNN evaluation under test-time graph distribution shifts, there are no established baseline methods for making direct comparisons. Therefore, we evaluate our proposed method by comparing it to several convolutional neural network (CNN) model evaluation methods applied to image data, specifically tailored to online evaluation scenarios. Note that existing CNN model evaluation methods are hard to directly apply to GNNs, since GNNs have entirely different convolutional architectures working on different data types. Therefore, we make necessary adaptations to enable these methods to work on graph-structured data for online GNN model evaluation. Specifically, we compare seven baseline methods in four categories, including *averaged confidence score (ConfScore)* (Hendrycks & Gimpel, 2016), *Entropy* (Guillory et al., 2021), *average thresholded confidence (ATC) score* (Garg et al., 2022) with its two variants ATC-NE (negative entropy) and ATC-MC (maximum confidence), as well as *threshold-based method* (Deng & Zheng, 2021) with three threshold values for Thres. ($\tau = \{0.7, 0.8, 0.9\}$). More details of these baseline methods can be found in Appendix D.

## 3.2 ONLINE GNN MODEL EVALUATION PERFORMANCE

In this section, we aim to answer **Q1** and report the results of LEBED in evaluating well-trained GNNs on the online node classification task under various graph distribution shifts in Table 1, Table 2, and Table 3. In general, it can be observed that our proposed LEBED achieves consistent best performance on evaluating various well-trained GNNs under all node feature shifts, domain shifts, and temporal shifts, showcasing strong positive correlations $\rho$ and outstanding $R^2$ fitting. This significantly demonstrates the effectiveness of our approach in capturing the correlation between LEBED and ground-truth test errors. For instance, for node feature shifts, our LEBED has $\rho = 0.90$ on Amazon-Photo for well-trained GCN online evaluation, significantly exceeding other comparison methods. Besides, many negative correlations are observed in ATC and Threshold-based methods, contrary to the expected positive correlations, underscoring the limitations of straightforwardly adapting existing methods for online GNN evaluation. For example, the ATC-MC method exhibits predominantly negative correlations when applied to online GNN evaluation in the context of domain shifts on ACMv9. However, it demonstrates positive correlations when assessing GCN on Amazon-Photo under feature shifts. The underlying reason could be that such adaptation fails to consider the distinctive neighbor aggregation mechanisms of GNNs intrinsic to the inherent structural characteristics of graph data. These inconsistent statistical correlations with ground truth test errors suggest that it is not a reliable metric for predicting generalization errors in online evaluation scenarios. Furthermore, our proposed LEBED demonstrates generally better performance under node feature shifts when compared to domain shifts and temporal shifts. For example, it achieves an average $\rho = 0.76$ on Cora across all GNNs, whereas it reaches $\rho = 0.52$ on both Citationv2 and OGB-arxiv. This discrepancy can be primarily attributed to the increased complexity associated with domain shifts and temporal shifts, where multi-faceted distribution variances impact various aspects of node features, graph structure, and scales.

Table 2: Evaluation performance of well-trained GNNs on ACMv9, DBLPv8, Citationv2 under **domain shifts** with Spearman rank correlation $\rho$ and linear fitting $R^2$. The best results are in bold.

| ACMv9 | GCN | | GAT | | SAGE | | GIN | | MLP | | avg. GNNs | |
|---|---|---|---|---|---|---|---|---|---|---|---|---|
| | $\rho$ | $R^2$ | $\rho$ | $R^2$ | $\rho$ | $R^2$ | $\rho$ | $R^2$ | $\rho$ | $R^2$ | $\rho$ | $R^2$ |
| ConfScore | 0.48 | 0.147 | 0.49 | 0.253 | 0.38 | 0.252 | -0.06 | 0.000 | 0.36 | 0.316 | 0.33 | 0.194 |
| Entropy | 0.49 | 0.177 | 0.48 | 0.271 | 0.38 | 0.285 | -0.07 | 0.000 | 0.36 | 0.364 | 0.33 | 0.219 |
| ATC-MC | -0.48 | 0.117 | -0.49 | 0.224 | -0.37 | 0.140 | -0.03 | 0.003 | -0.35 | 0.214 | -0.34 | 0.140 |
| ATC-NE | -0.48 | 0.144 | -0.48 | 0.245 | -0.38 | 0.138 | -0.04 | 0.004 | -0.35 | 0.208 | -0.35 | 0.148 |
| Thres. ($\tau = 0.7$) | -0.47 | 0.132 | -0.49 | 0.228 | -0.38 | 0.267 | 0.06 | 0.000 | -0.36 | 0.303 | -0.33 | 0.186 |
| Thres. ($\tau = 0.8$) | -0.47 | 0.165 | -0.49 | 0.280 | -0.39 | 0.305 | 0.09 | 0.002 | -0.37 | 0.370 | -0.33 | 0.224 |
| Thres. ($\tau = 0.9$) | -0.48 | 0.222 | -0.48 | 0.344 | -0.40 | 0.337 | 0.11 | 0.006 | -0.37 | **0.437** | -0.32 | 0.269 |
| **LeBed (Ours)** | **0.52** | **0.434** | **0.52** | **0.540** | **0.61** | **0.501** | **0.46** | **0.176** | **0.47** | 0.405 | **0.52** | **0.411** |

| DBLPv8 | GCN | | GAT | | SAGE | | GIN | | MLP | | avg. GNNs | |
|---|---|---|---|---|---|---|---|---|---|---|---|---|
| | $\rho$ | $R^2$ | $\rho$ | $R^2$ | $\rho$ | $R^2$ | $\rho$ | $R^2$ | $\rho$ | $R^2$ | $\rho$ | $R^2$ |
| ConfScore | 0.34 | 0.292 | 0.39 | 0.391 | 0.23 | 0.426 | 0.19 | 0.213 | 0.23 | 0.515 | 0.28 | 0.367 |
| Entropy | 0.37 | 0.369 | 0.42 | 0.468 | 0.23 | 0.475 | 0.20 | 0.226 | 0.23 | 0.580 | 0.29 | 0.424 |
| ATC-MC | -0.31 | 0.217 | -0.37 | 0.307 | -0.26 | 0.333 | -0.22 | 0.179 | -0.23 | 0.286 | -0.28 | 0.265 |
| ATC-NE | -0.34 | 0.275 | -0.41 | 0.353 | -0.27 | 0.324 | -0.24 | 0.171 | -0.22 | 0.267 | -0.30 | 0.278 |
| Thres. ($\tau = 0.7$) | -0.31 | 0.271 | -0.36 | 0.395 | -0.22 | 0.445 | -0.20 | 0.220 | -0.27 | 0.575 | -0.27 | 0.381 |
| Thres. ($\tau = 0.8$) | -0.36 | 0.365 | -0.38 | 0.477 | -0.20 | 0.473 | -0.20 | 0.236 | -0.29 | 0.630 | -0.29 | 0.436 |
| Thres. ($\tau = 0.9$) | -0.40 | 0.485 | -0.42 | **0.568** | -0.19 | 0.521 | -0.20 | 0.249 | -0.31 | **0.669** | -0.30 | 0.498 |
| **LeBed (Ours)** | **0.79** | **0.784** | **0.82** | 0.276 | **0.83** | **0.779** | **0.79** | **0.763** | **0.60** | 0.551 | **0.76** | **0.615** |

| Citationv2 | GCN | | GAT | | SAGE | | GIN | | MLP | | avg. GNNs | |
|---|---|---|---|---|---|---|---|---|---|---|---|---|
| | $\rho$ | $R^2$ | $\rho$ | $R^2$ | $\rho$ | $R^2$ | $\rho$ | $R^2$ | $\rho$ | $R^2$ | $\rho$ | $R^2$ |
| ConfScore | 0.39 | 0.049 | 0.44 | 0.095 | 0.26 | 0.131 | -0.02 | 0.002 | 0.30 | 0.316 | 0.27 | 0.119 |
| Entropy | 0.40 | 0.077 | 0.45 | 0.138 | 0.29 | 0.177 | -0.02 | 0.004 | 0.32 | 0.342 | 0.29 | 0.148 |
| ATC-MC | -0.41 | 0.041 | -0.42 | 0.055 | -0.28 | 0.038 | -0.07 | 0.003 | -0.30 | 0.099 | -0.30 | 0.047 |
| ATC-NE | -0.43 | 0.061 | -0.47 | 0.081 | -0.29 | 0.035 | -0.07 | 0.004 | -0.34 | 0.093 | -0.32 | 0.055 |
| Thres. ($\tau = 0.7$) | -0.37 | 0.044 | -0.42 | 0.097 | -0.27 | 0.171 | 0.02 | 0.003 | -0.70 | 0.333 | -0.35 | 0.129 |
| Thres. ($\tau = 0.8$) | -0.35 | 0.053 | -0.43 | 0.125 | -0.31 | 0.197 | 0.04 | 0.004 | -0.70 | 0.307 | -0.35 | 0.137 |
| Thres. ($\tau = 0.9$) | -0.36 | 0.071 | -0.45 | 0.167 | -0.37 | 0.210 | 0.06 | 0.002 | -0.68 | 0.272 | -0.36 | 0.144 |
| **LeBed (Ours)** | **0.63** | **0.200** | **0.53** | **0.228** | **0.45** | **0.275** | **0.51** | **0.210** | **0.47** | **0.356** | **0.52** | **0.254** |

Table 3: Evaluation performance of well-trained GNNs on OGB-arxiv under **temporal shifts** with Spearman rank correlation $\rho$ and linear fitting $R^2$. The best results are in bold.

| OGB-arxiv | GCN | | GAT | | SAGE | | GIN | | MLP | | avg. GNNs | |
|---|---|---|---|---|---|---|---|---|---|---|---|---|
| | $\rho$ | $R^2$ | $\rho$ | $R^2$ | $\rho$ | $R^2$ | $\rho$ | $R^2$ | $\rho$ | $R^2$ | $\rho$ | $R^2$ |
| ConfScore | 0.09 | 0.117 | 0.09 | 0.109 | 0.21 | 0.226 | 0.51 | 0.362 | 0.52 | 0.551 | 0.28 | 0.273 |
| Entropy | 0.12 | 0.145 | 0.10 | 0.110 | 0.26 | 0.242 | 0.52 | 0.386 | 0.62 | 0.643 | 0.32 | 0.305 |
| ATC-MC | -0.11 | 0.110 | -0.11 | 0.094 | -0.21 | 0.222 | -0.51 | 0.366 | -0.58 | 0.635 | -0.30 | 0.286 |
| ATC-NE | -0.10 | 0.134 | -0.11 | 0.096 | -0.26 | **0.257** | -0.54 | **0.392** | -0.64 | **0.754** | -0.33 | **0.327** |
| Thres. ($\tau = 0.7$) | -0.07 | 0.124 | -0.08 | 0.144 | -0.19 | 0.231 | -0.51 | 0.356 | -0.43 | 0.484 | -0.26 | 0.268 |
| Thres. ($\tau = 0.8$) | -0.05 | 0.133 | -0.06 | 0.177 | -0.17 | 0.237 | -0.50 | 0.348 | -0.38 | 0.426 | -0.23 | 0.264 |
| Thres. ($\tau = 0.9$) | -0.04 | 0.146 | -0.05 | **0.219** | -0.14 | 0.245 | -0.50 | 0.339 | -0.28 | 0.364 | -0.20 | 0.262 |
| **LeBed (Ours)** | **0.51** | **0.217** | **0.53** | 0.185 | **0.36** | 0.093 | **0.54** | 0.280 | **0.87** | 0.661 | **0.56** | 0.287 |

Table 4: Ablation study on with (w/) and without (w/o) the proposed $D_{\text{stru.}}$ based parameter-free optimality criterion on ACMv9 under test-time domain shifts.

| ACMv9 | GCN | | GAT | | SAGE | | GIN | | MLP | |
|---|---|---|---|---|---|---|---|---|---|---|
| Iters. $[q]$ (w/o $D_{\text{stru.}}$) | $\rho$ | $R^2$ | $\rho$ | $R^2$ | $\rho$ | $R^2$ | $\rho$ | $R^2$ | $\rho$ | $R^2$ |
| 20 | 0.27 | 0.204 | 0.46 | 0.600 | 0.26 | 0.119 | 0.35 | 0.170 | 0.44 | 0.389 |
| 40 | 0.17 | 0.056 | 0.46 | **0.603** | -0.10 | 0.028 | 0.25 | 0.118 | 0.42 | 0.375 |
| 80 | 0.30 | 0.136 | 0.46 | 0.591 | -0.45 | 0.140 | -0.22 | 0.033 | 0.43 | 0.352 |
| 160 | 0.24 | 0.068 | 0.46 | 0.559 | -0.53 | 0.170 | -0.43 | 0.125 | 0.40 | 0.281 |
| 200 | 0.13 | 0.026 | 0.46 | 0.535 | -0.55 | 0.178 | -0.46 | 0.140 | 0.42 | 0.249 |
| **LeBed** (w/ $D_{\text{stru.}}$) | **0.52** | **0.434** | **0.52** | 0.540 | **0.61** | **0.501** | **0.46** | **0.176** | **0.47** | **0.405** |

## 3.3 IN-DEPTH ANALYSIS OF THE PROPOSED LEBED

**Ablation Study of Parameter-free Optimality Criterion.** In Table 4, we compare the performance with and without the proposed $D_{\text{stru.}}$ based criterion to verify its effectiveness on ACMv9 dataset under test-time domain shift (**Q2**). As can be observed, compared with using fixed $q$ iterations, our proposed parameter-free $D_{\text{stru.}}$ criterion performs a better indicator to reflect the optimality of GNN re-training, facilitating more accurate optimal weight parameter distance calculation for LeBed. More results on the optimal iteration steps for all GNNs on all datasets, along with the time complexity analysis, are provided in Appendix C. The dominant factor affecting the time complexity is the number of re-training iterations $q$. This reflects the necessity and effectiveness of the proposed

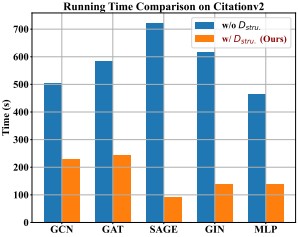 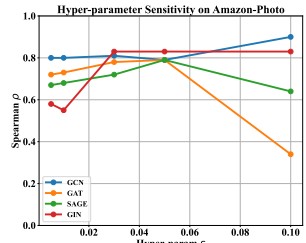 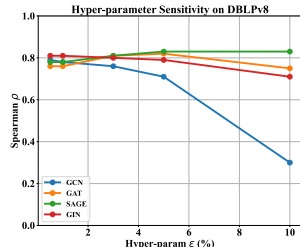

Figure 3: Running time comparison on Citationv2 dataset w/ and w/o the proposed $D_{\text{stru.}}$ based criterion.

Figure 4: Hyper-parameter sensitivity analysis on $\epsilon$ in the proposed parameter-free optimality criterion. (*left*: Amazon-Photo dataset with fixed constant setting; *right*: DBLPv8 dataset with fixed ratio (%) setting.)

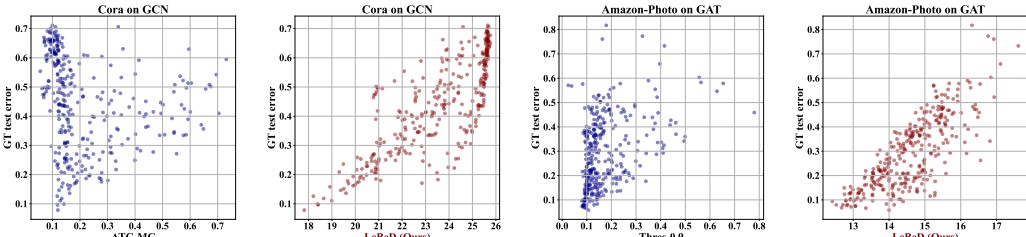

Figure 5: Correlation visualization for ATC-MC of GCN on Cora, our LEBED of GCN on Cora, Thres.($\tau = 0.9$) of GAT on Amazon-Photo, and our LEBED of GAT on Amazon-Photo.

$D_{\text{stru.}}$ based parameter-free optimality criterion through reducing $q$. The running time comparison on Citationv2 in seconds is shown in Fig. 3 with a single GeForce RTX 3080 GPU and 200 iterations for w/ $D_{\text{stru.}}$. It can be seen that the proposed criterion could decrease the running time significantly, demonstrating its great efficiency for online GNN evaluation.

**Hyper-parameter Sensitivity Analysis.** We analyze the hyper-parameter sensitivity of $\epsilon$ on the proposed method for answering **Q3**, and the results on various well-trained GNN models can be seen in Fig. 4. We consider two different types of strategies to set the $\epsilon$, *i.e.*, fixed constant setting and fixed ratio setting, where the former sets $\epsilon$ as a fixed constant for all test graphs on each to-be-evaluated GNN, *e.g.*, 0.02 denotes constant distance, and the latter sets $\epsilon$ as a fixed rations of differences between two terms in Eq. (6), *e.g.*, 3% discrepancy tolerance. The smaller the $\epsilon$, the less discrepancy tolerance is allowed, the performance would be better. The experimental results show that under certain ranges with different hyper-parameter setting strategies, the proposed LEBED could achieve relatively consistent performance with low hyper-parameter sensitivity to $\epsilon$.

**Correlation Visualization.** In Fig. 5, we visualize the correlation relationship between the ground-truth test errors and our proposed LEBED under node feature shift datasets, and make comparisons with existing methods on well-trained GCN and GAT for answering **Q4**. As can be observed, our proposed LEBED achieves stronger correlations compared with other baselines, verifying its effectiveness for associating with ground-truth test errors for online GNN evaluation.

## 4 CONCLUSION

In this work, we have studied a new problem, online GNN evaluation, with an effective learning behavior discrepancy score, dubbed LEBED, to estimate the test-time generalization errors of well-trained GNNs on real-world graphs under test-time distribution shifts. A novel GNN re-training strategy with a parameter-free optimality criterion comprehensively incorporates both node prediction discrepancy and structure reconstruction discrepancy to precisely compute LEBED. Extensive experiments on real-world unlabeled graphs under diverse distribution shifts could verify the effectiveness of the proposed method. Our method assumes that the class label space is unchanged across training and testing graphs though covariate shifts may exist between the two. We will look into relaxing this assumption and address a broader range of more complex real-world graph data distribution shifts in the future.

## ACKNOWLEDGEMENT

S. Pan was supported in part by the Australian Research Council (ARC) under grants FT210100097 and DP240101547, and the CSIRO – National Science Foundation (US) AI Research Collaboration Program.

## ETHICS STATEMENT

Our research is solely focused on tackling scientific research questions and does not entail the participation of human subjects, animals, or environmentally sensitive materials. As a result, we anticipate no ethical concerns or conflicts of interest in our study. We are committed to upholding the utmost standards of scientific integrity and ethical practice throughout our research, thereby guaranteeing the accuracy and dependability of our results.

## REPRODUCIBILITY STATEMENT

We have carefully provided a clear and comprehensive formalization of our proposed method in the main submission. Additionally, we delve into the dataset information and method implementation details in Appendix B and D to ensure reproducibility.

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

APPENDIX

This is the appendix of the work: **'Online GNN Evaluation Under Test-time Graph Distribution Shifts'**. In this appendix, we present more additional method details, experimental results, and discussions regarding the proposed *online GNN evaluation* problem and the proposed LEBED score method.

## A  RELATED WORK

**Predicting Model Generalization Error.** Our work is relevant to the line of research on predicting model generalization error, which aims to develop a good estimator of a source-domain well-trained model's performance on unlabeled data from the unknown distributions in the target domain (Deng & Zheng, 2021; Garg et al., 2022; Yu et al., 2022; Deng et al., 2022; Guillory et al., 2021; Deng et al., 2021). Typically, Hendrycks et al. (Hendrycks & Gimpel, 2016) and Guillory et al. (Guillory et al., 2021) proposed to estimate a convolutional neural network (CNN) classifier's performance on image data with delicately designed metrics, *i.e.*, averaged confidence score (ConfScore), averaged entropy (Entropy) score, and difference of confidences (DoC) score, to estimate and reflect the model accuracy by means of its classifier's softmax outputs. Garg et al. (Garg et al., 2022) also proposed to learn a softmax probability based average thresholded confidence (ATC) score by estimating the fraction of unlabeled images above the threshold calculated by the validation dataset. In contrast, Deng et al. (Deng & Zheng, 2021; Deng et al., 2021) directly predicted CNN classifier accuracy by deriving data distribution distance features between the training and test images with a linear regression model. Compared with directly estimating the accuracy with the image data-level discrepancy, Yu et al. (Yu et al., 2022) proposed to calculate the well-trained CNN models parameter space difference as projection norm, and took it as an estimation of out-of-distribution.

Note that, these existing methods mostly focus on evaluating CNN model classifiers on image data in computer vision, and the formulation of evaluating GNNs for graph structural data still remains under-explored in graph machine learning. Concretely, conducting online GNN evaluation for graph structural data at test time has two critical challenges: (1) different from Euclidean image data, graph structural data lies in non-Euclidean space with complex and non-independent node-edge interconnections, so that its node contexts and structural characteristics significantly vary under wide-range graph data distributions, posing severe challenges for GNN model evaluation when serving on unknown graph data distributions. (2) GNNs have entirely different convolutional architectures from those of CNNs, when GNN convolution aggregates neighbor node messages along graph structures. Such that GNNs trained on the observed training graph might well fit its graph structure, and due to the complexity and diversity of graph structures, serving well-trained GNNs on unlabeled test distributions that they have not encountered before would incur more performance uncertainty. Furthermore, in online GNN deployment scenarios where access to the training graph is restricted, the feasibility of measuring data-level discrepancies becomes unattainable.

Hence, in this work, we first investigate the online GNN evaluation problem for graph structural data with a clear problem definition and a feasible solution, taking a significant step towards understanding and evaluating GNN models for practical online GNN deployment and serving. Besides, our proposed LEBED can be taken as an extension of the work (Yu et al., 2022) to GNN models on graph structural data. Differently, our method makes effective use of additional self-supervision signals derived from graph structures, thanks to the incorporation of the proposed parameter-free optimality criterion, which clearly distinguishes our method from Yu et al. (2022).

**Unsupervised Graph Domain Adaption.** Our work is also relevant to the unsupervised graph domain adaption (UGDA) problem, whose goal is to develop a GNN model with both labeled source graphs and unlabeled target graphs for better generalization ability on target graphs. Typically, existing UGDA methods focus on mitigating the domain gap by aligning the source graph distribution with the target one. For instance, Yang et al. (Yang et al., 2021) and Shen et al. (Shen et al., 2020) optimized domain distance loss based on the statistical information of datasets, *e.g.*, maximum mean discrepancy (MMD) metric. Moreover, DANE (Zhang et al., 2019) and UDAGCN (Wu et al., 2020) introduced domain adversarial methods to learn domain-invariant embeddings across the source domain and the target domain. Therefore, the critical distinction between our work and UGDA is that our work is primarily concerned with evaluating the GNNs' performance on unseen graphs with-

Table A1: Dataset statistics with various test-time graph data distribution shifts.

| Distribution shifts | Datasets | #Nodes | #Edges | #Classes | #Train/Val/Test Graphs (Adapted) |
|---|---|---|---|---|---|
| Node feature shifts | Cora (Yang et al., 2016) | 2,703 | 5,278 | 10 | 1/1/320 (8-artifices) |
| | Amazon-Photo (Shchur et al., 2018) | 7,650 | 119,081 | 10 | 1/1/320 (8-artifices) |
| Domain shifts | ACMv9 (Wu et al., 2020) | 7,410 | 22,270 | 6 | 1/1/369 (3-domains: A, D, C) |
| | DBLPv8 (Wu et al., 2020) | 5,578 | 14,682 | 6 | 1/1/369 (3-domains: A, D, C) |
| | Citationv2 (Wu et al., 2020) | 4,715 | 13,466 | 6 | 1/1/369 (3-domains: A, D, C) |
| Temporal shifts | OGB-arxiv (Pareja et al., 2020) | 169,343 | 1,166,243 | 40 | 1/1/360 (3-periods) |

Table A2: Details of test graphs under diverse graph distribution shifts with adapted expansions.

| Distribution shift | Datasets | # Raw TestGs | # Feat Perturbs | # Feat Masks | # SubGs | # Edge Perturbs | Totals |
|---|---|---|---|---|---|---|---|
| Node feature shifts | Cora | 8 | 20 | 20 | - | - | 320 |
| | Amazon-Photo | 8 | 20 | 20 | - | - | 320 |
| Domain shifts | ACMv9 | 1 | 10 | 10 | 10 | 10 | A: 41, D: 164, C: 164 (369) |
| | DBLPv8 | 1 | 10 | 10 | 10 | 10 | D: 41, A: 164, C: 164 (369) |
| | Citationv2 | 1 | 10 | 10 | 10 | 10 | C: 41, D: 164, A: 164 (369) |
| Temporal shifts | OGB-arxiv | 3 | 30 | 30 | 30 | 30 | 360 |

out labels under test-time distribution shifts, rather than improving the GNNs' generalization ability when adapting to unlabeled target graphs.

**Graph Out-of-distribution (OOD) Generalization.** Out-of-distribution (OOD) generalization (Li et al., 2022; Zhu et al., 2021) on graphs aims to develop a GNN model given several different but related source domains, so that the developed model can generalize well to unseen target domains. Li et al. (Li et al., 2022) categorized existing graph OOD generalization methodologies into three conceptually different branches, *i.e.*, data, model, and learning strategy, based on their positions in the graph machine learning pipeline. We would like to highlight that, even ODD generalization and our proposed online GNN evaluation both pay attention to the general graph data distribution shift issue, we have different purposes: our proposed online GNN evaluation aims to evaluate well-trained GNNs' performance on unseen test graphs, while graph ODD generalization aims to develop a new GNN model for improve its performance or generalization capability on unseen test graphs.

# B    TEST-TIME DATASET DETAILS

The dataset statistics and details of the adapted test graphs for all datasets under various distribution shifts are provided in Table A1 and Table A2, respectively. In general, we expand all raw test graphs with four types of strategies, including feature perturbation with Gaussian covariate shifts, feature masking, sub-graph sampling, and edge drop, with the default random range $[0.1, 0.7]$ in uniform distribution sampling with different quantities.

More specifically, for node feature shifts, we use the artificial transformations of Cora with GAT-generation, and Amazon with GCN-generation according to the work (Wu et al., 2022). Each dataset has eight raw test graphs and we only impose feature-level expansion strategies on them. For domain shifts, we consider full/train/val/test graphs for each domain under the inductive setting, and each is expanded with all four strategies. For each domain shift case, *e.g.*, ACMv9, we use the training graph for well-training GNN models, and its online test set contains its test graph expansions, and all other domains' (DBLPv8 and Citationv2) all graph expansions for A→(A, D, C) cross-domain evaluation. For temporal shifts, we follow Wu et al. (2022) using the year range $[1950, 2011]$ corresponded sub-graph as the training graph, $[2011, 2014]$ for validation, and remain three year periods $[2014, 2016]$, $[2016, 2018]$, $[2018, 2020]$ as the raw test graphs involving all four-type expansion strategies.

Table A3: Hyper-parameter sensitivity analysis on $\epsilon$ in the proposed parameter-free optimality criterion on Amazon-Photo dataset with the fixed constant setting.

| Amazon-Photo | GCN | | GAT | | SAGE | | GIN | |
|---|---|---|---|---|---|---|---|---|
| $\epsilon$ | $\rho$ | $R^2$ | $\rho$ | $R^2$ | $\rho$ | $R^2$ | $\rho$ | $R^2$ |
| 0.005 | 0.80 | 0.723 | 0.72 | 0.550 | 0.67 | 0.478 | 0.58 | 0.238 |
| 0.01 | 0.80 | 0.724 | 0.73 | 0.560 | 0.68 | 0.467 | 0.55 | 0.037 |
| 0.03 | 0.81 | 0.670 | 0.78 | 0.614 | 0.72 | 0.365 | 0.83 | 0.705 |
| 0.05 | 0.79 | 0.566 | 0.79 | 0.591 | 0.79 | 0.425 | 0.83 | 0.705 |
| 0.1 | 0.90 | 0.730 | 0.34 | 0.052 | 0.64 | 0.591 | 0.83 | 0.705 |
| 0.2 | 0.90 | 0.730 | 0.54 | 0.338 | 0.63 | 0.586 | 0.83 | 0.705 |

Table A4: Hyper-parameter sensitivity analysis on $\epsilon$ in the proposed parameter-free optimality criterion on DBLPv8 dataset with the fixed ratio setting.

| DBLPv8 | GCN | | GAT | | SAGE | | GIN | |
|---|---|---|---|---|---|---|---|---|
| $\epsilon(\%)$ | $\rho$ | $R^2$ | $\rho$ | $R^2$ | $\rho$ | $R^2$ | $\rho$ | $R^2$ |
| 0.5 | 0.79 | 0.784 | 0.76 | 0.607 | 0.78 | 0.604 | 0.81 | 0.650 |
| 1 | 0.78 | 0.768 | 0.76 | 0.568 | 0.78 | 0.601 | 0.81 | 0.672 |
| 3 | 0.76 | 0.706 | 0.81 | 0.430 | 0.81 | 0.623 | 0.80 | 0.727 |
| 5 | 0.71 | 0.581 | 0.82 | 0.276 | 0.83 | 0.687 | 0.79 | 0.763 |
| 10 | 0.30 | 0.120 | 0.75 | 0.296 | 0.83 | 0.779 | 0.71 | 0.796 |
| 20 | 0.77 | 0.798 | 0.72 | 0.767 | 0.76 | 0.737 | 0.48 | 0.529 |

## C  IN-DEPTH ANALYSIS AND MORE RESULTS

### C.1  TIME COMPLEXITY ANALYSIS.

The time complexity of the proposed LEBED can be analyzed along with its learning procedure. Taking $L$-layer GCN as the example, for S1 test graph inference, there is one-time computation complexity approximately $\mathcal{O}(LMd^2) + \mathcal{O}(E)$, where $E$ is the number of edges on the test graph $G_{te}$, and we shorten it to $\mathcal{O}(\Omega) = \mathcal{O}(LMd^2) + \mathcal{O}(E)$. For S2 GNN re-training, the time complexity can be $\mathcal{O}(q * \Omega)$ highly related to the number of iterations $q$. Then, the parameter-free optimality criterion takes $\mathcal{O}(M^2)$. For S3 LEBED score computation, the time complexity can be about $\mathcal{O}(w)$ where $w$ is the number of GNN weight parameters. Overall, the time complexity of the proposed LEBED can be approximated as $\mathcal{O}(\Omega) + \mathcal{O}(q * \Omega) + \mathcal{O}(M^2) + \mathcal{O}(w)$, and the dominant factors affecting the time complexity are the number of re-training iterations $q$ and the size of the graph $M$. This reflects the necessity and effectiveness of the proposed $D_{\text{stru.}}$ based parameter-free optimality criterion through reducing $q$.

### C.2  MORE RESULTS ON THE PROPOSED PARAMETER-FREE OPTIMALITY CRITERION

We provide wider range $\epsilon$ hyper-parameter results for both fixed constant setting in Table A3 and fixed ration setting in Table A4. It can be generally observed that $\epsilon$ could affect the online evaluation performance but it shows relatively low hyper-parameter sensitivity in certain ranges.

Moreover, we present the average iteration stop epochs determined by the proposed $D_{\text{stru.}}$ parameter-free criterion, which indicate the point at which the re-trained GNN achieves optimality in Table A5. It can be generally observed that our proposed criterion could constrain the GNN re-training process with $D_{\text{stru.}}$ self-supervision signals to stop the over-retraining at early stages. Besides, more correlation visualization comparison results of different well-trained GNNs under diverse graph distribution shifts are shown in Fig. A6, A7, A8, and A9.

Table A5: Average optimal iterations produced by the proposed $D_{\text{stru.}}$ based parameter-free criterion.

| # Avg. Opt. Iters. | Cora | Amazon-Photo | ACMv9 | DBLPv8 | Citationv2 | OGB-arxiv |
|---|---|---|---|---|---|---|
| GCN | 79 | 5 | 12 | 175 | 42 | 585 |
| GAT | 46 | 420 | 188 | 30 | 32 | 157 |
| SAGE | 73 | 166 | 7 | 161 | 14 | 377 |
| GIN | 110 | 5 | 23 | 131 | 33 | 197 |
| MLP | 9 | 254 | 13 | 57 | 20 | 344 |

---

**Algorithm 1** Learning Behavior Discrepancy (LEBED) Score Computation.

---

**Require:** Real-world unseen and unlabeled test graph $G_{\text{te}}$, online-deployed GNN model $\text{GNN}_{\boldsymbol{\theta}^*_{\text{tr}}}$ that has been well-trained with known initialization parameters $\boldsymbol{\theta}_0$, number of re-training iterations $q$, structure reconstruction discrepancy hyper-parameter $\epsilon$.

**Ensure:** LEBED score.

1: Input $G_{\text{te}}$ to well-trained $\text{GNN}_{\boldsymbol{\theta}^*_{\text{tr}}}$ to acquire online-generated node representations $\mathbf{Z}^*_{\text{te}}$ and pseudo labels $\mathbf{Y}^*_{\text{te}}$ according to Eq. (4);

2: Let $\boldsymbol{\theta}_{\text{te}} = \boldsymbol{\theta}_0$;

3: **while** $0 < t <= q$ **do**

4:     Re-train $\text{GNN}_{\boldsymbol{\theta}_{\text{te}}}$ with the prediction discrepancy $D_{\text{Pred.}}$ according to Eq. (5), where $\boldsymbol{\theta}_{\text{te},t+1} \leftarrow \boldsymbol{\theta}_{\text{te},t} - \nabla_{\boldsymbol{\theta}_{\text{te}}} \mathcal{L}_{\text{cls}} [\text{GNN}_{\boldsymbol{\theta}_{\text{te}}} (G_{\text{te}})]$;

5:     Compute the parameter-free structure reconstruction discrepancy criterion $D_{\text{Stru.}}$ according to Eq. 6 and Eq. 7;

6:     **if** $D_{\text{Stru.}} < \epsilon$ **then**

7:         $\boldsymbol{\theta}^\dagger_{\text{te}} \leftarrow \boldsymbol{\theta}_{\text{te},t}$ and break;

8:     **end if**

9: **end while**

10: Until $\boldsymbol{\theta}^\dagger_{\text{te}} \leftarrow \boldsymbol{\theta}_{\text{te},q}$;

11: Calculate the proposed LEBED score according to Eq. (8)

---

# D IMPLEMENTATION DETAILS

## D.1 OVERALL ALGORITHM AND IMPLEMENTATIONS

The overall procedure of the proposed method is provided in Algorithm 1. In our experiments, we use Pytorch geometric library (Fey & Lenssen, 2019) and four GeForce RTX 3080 GPUs for all implementations.

## D.2 BASELINE METHODS DETAILS

We consider the baseline methods following Yu et al. (2022) and Deng & Zheng (2021). For adapting to online GNN evaluation, we only compare with baselines that do not require accessing the original training graph when estimating the generalization errors on the test graph.

**ConfScore.** This metric (Hendrycks & Gimpel, 2016) utilizes the softmax outputs of classifiers of the well-trained CNNs on the unseen and unlabeled test images, which can be calculated as:

$$\text{ConfScore} = \frac{1}{M} \sum_{j=1}^{M} \max_k \text{Softmax} \left( f_{\boldsymbol{\theta}^*} \left( \mathbf{x}'_j \right) \right)_k, \tag{9}$$

where $f_{\boldsymbol{\theta}^*}(\cdot)$ denotes the well-trained CNN's classifier with the optimal parameter $\boldsymbol{\theta}^*$, and $s(\cdot)$ denotes the score function working on the softmax prediction of $f_{\boldsymbol{\theta}^*}(\cdot)$. When the context is clear, we will use $f(\cdot)$ for simplification.

**Entropy** This metric (Hendrycks & Gimpel, 2016) calculates the entropy of the softmax outputs of the well-trained CNN classifiers as:

$$\text{Entropy} = \frac{1}{M} \sum_{j=1}^{M} \text{Ent} \left( \text{Softmax} \left( f_{\boldsymbol{\theta}^*} \left( \mathbf{x}'_j \right) \right) \right), \tag{10}$$

where $\text{Ent}(f(\mathbf{x}'_j)) = -\sum_k f_k(\mathbf{x}'_j) \log\left(f_k(\mathbf{x}'_j)\right)$.

**Average Thresholded Confidence (ATC) & Its Variants.** This metric (Garg et al., 2022) learns a threshold on CNN's confidence to estimate the accuracy as the fraction of unlabeled images whose confidence scores exceed the threshold as:

$$\text{ATC} = \frac{1}{M} \sum_{j=1}^{M} \mathbf{1}\left\{s\left(\text{Softmax}\left(f_{\boldsymbol{\theta}^*}\left(\mathbf{x}'_j\right)\right)\right) < t\right\}, \tag{11}$$

We adopted two different score functions, deriving two variants as: (1) Maximum confidence variant ATC-MC with $s(f(\mathbf{x}'_j)) = \max_{k \in \mathcal{Y}} f_k(\mathbf{x}'_j)$; and (2) Negative entropy variant ATC-NE with $s(f(\mathbf{x}'_j)) = \sum_k f_k(\mathbf{x}'_j) \log\left(f_k(\mathbf{x}'_j)\right)$, where $\mathcal{Y} = \{1, 2, \ldots, C\}$ is the label space. And for $t$ in Eq. (11), it is calculated based on the validation set of the observed training set $(\mathbf{x}^{\text{val}}_u, \mathbf{y}^{\text{val}}_u) \in \mathcal{S}_{\text{val}}$:

$$\frac{1}{N_{\text{val}}} \sum_{u=1}^{N_{\text{val}}} \mathbf{1}\left\{s\left(\text{Softmax}\left(f_{\boldsymbol{\theta}^*}\left(\mathbf{x}^{\text{val}}_u\right)\right)\right) < t\right\} = \frac{1}{N_{\text{val}}} \sum_{u=1}^{N_{\text{val}}} \mathbf{1}\left\{p\left(\mathbf{x}^{\text{val}}_u; \boldsymbol{\theta}^*\right) \neq \mathbf{y}^{\text{val}}_u\right\}, \tag{12}$$

where $p(\cdot)$ denotes the predicted labels of samples.

**Threshold-based Method.** This is an intuitive solution introduced by (Deng & Zheng, 2021), which is not a learning-based method. It follows the basic assumption that a class prediction will likely be correct when it has a high softmax output score. Then, the threshold-based method would provide the estimated accuracy of a model as:

$$\text{Test Error} = 1 - \frac{\sum_{i=1}^{M} \mathbf{1}\left\{\max\left(f_{\boldsymbol{\theta}^*}(\mathbf{x}'_j)\right) > \tau\right\}}{M}, \tag{13}$$

where $\tau$ is the pre-defined thresholds as $\tau = \{0.7, 0.8, 0.9\}$ on the output softmax logits of CNNs. This metric calculates the percentage of images in the entire dataset whose maximum entries of logits are greater than the threshold $\tau$, which indicates these images are correctly classified.

### D.3 WELL-TRAINED GNNS DETAILS

We use two-layer GNN models for all experiments. Except for OGB-arxiv dataset with the hidden feature dimension 128 and the output embedding dimension 16, all other datasets are with hidden feature dimension 256 and the output embedding dimension 32. The well-trained GNNs performance on each dataset's validation set is shown in Table A6, and the ground-truth test error distributions for all datasets are provided in Fig. A1, A2, A3, A4, A5.

Table A6: GNN well-training hyper-parameter settings (LR: learning rate; WD: weight decay) and node classification accuracy $\text{ACC}_{\text{val}}$ (%) performance on validation set.

| Datasets | Cora | | | Amazon-Photo | | | OGB-arxiv | | |
|---|---|---|---|---|---|---|---|---|---|
| Models | LR | WD | $\text{ACC}_{\text{val}}$ (%) | LR | WD | $\text{ACC}_{\text{val}}$ (%) | LR | WD | $\text{ACC}_{\text{val}}$ (%) |
| GCN | 1.00E-03 | 5.00E-04 | 88.44 | 1.00E-03 | 0.00E+00 | 95.37 | 1.00E-02 | 1.00E-03 | 57.78 |
| GAT | 1.00E-03 | 5.00E-04 | 99.00 | 1.00E-03 | 0.00E+00 | 98.25 | 2.00E-02 | 5.00E-04 | 58.04 |
| SAGE | 1.00E-03 | 5.00E-04 | 99.45 | 1.00E-03 | 0.00E+00 | 99.88 | 1.00E-02 | 5.00E-04 | 59.35 |
| GIN | 5.00E-04 | 5.00E-04 | 89.29 | 5.00E-05 | 0.00E+00 | 85.22 | 1.00E-02 | 5.00E-04 | 53.28 |
| MLP | 1.00E-03 | 5.00E-04 | 99.78 | 1.00E-04 | 0.00E+00 | 99.54 | 1.00E-02 | 5.00E-03 | 60.45 |

| Datasets | ACMv9 | | | DBLPv8 | | | Citationv2 | | |
|---|---|---|---|---|---|---|---|---|---|
| Models | LR | WD | $\text{ACC}_{\text{val}}$ (%) | LR | WD | $\text{ACC}_{\text{val}}$ (%) | LR | WD | $\text{ACC}_{\text{val}}$ (%) |
| GCN | 1.00E-03 | 1.00E-05 | 76.65 | 1.00E-03 | 1.00E-04 | 75.94 | 1.00E-03 | 1.00E-04 | 81.95 |
| GAT | 2.00E-03 | 1.00E-05 | 75.17 | 2.00E-03 | 1.00E-05 | 71.99 | 5.00E-04 | 1.00E-04 | 81.10 |
| SAGE | 2.00E-03 | 1.00E-05 | 75.03 | 2.00E-03 | 5.00E-05 | 68.4 | 1.00E-03 | 1.00E-04 | 79.62 |
| GIN | 5.00E-04 | 5.00E-05 | 70.18 | 2.00E-03 | 1.00E-04 | 68.22 | 2.00E-03 | 5.00E-05 | 80.25 |
| MLP | 1.00E-04 | 1.00E-04 | 73.82 | 1.00E-04 | 1.00E-04 | 71.63 | 5.00E-04 | 1.00E-04 | 79.41 |

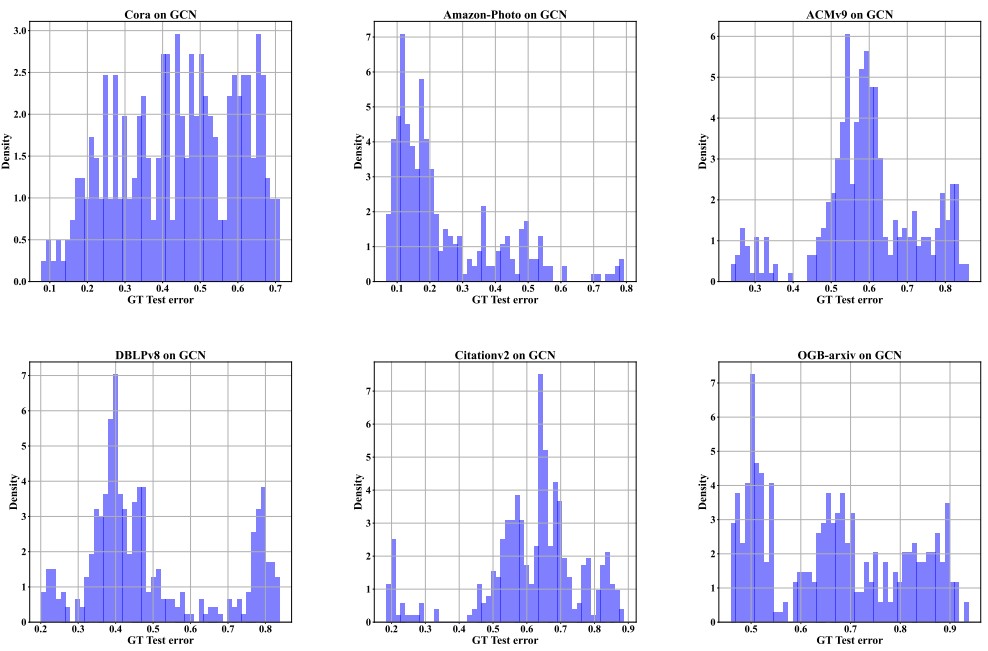

Figure A1: Ground-truth test error distributions of all test-time graphs on well-trained GCNs.

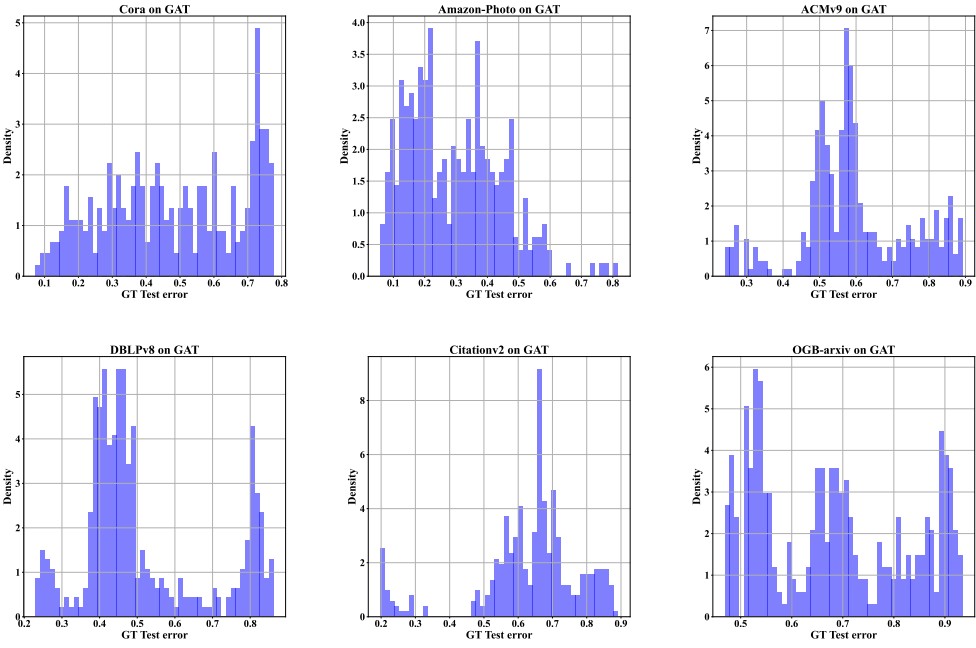

Figure A2: Ground-truth test error distributions of all test-time graphs on well-trained GATs.

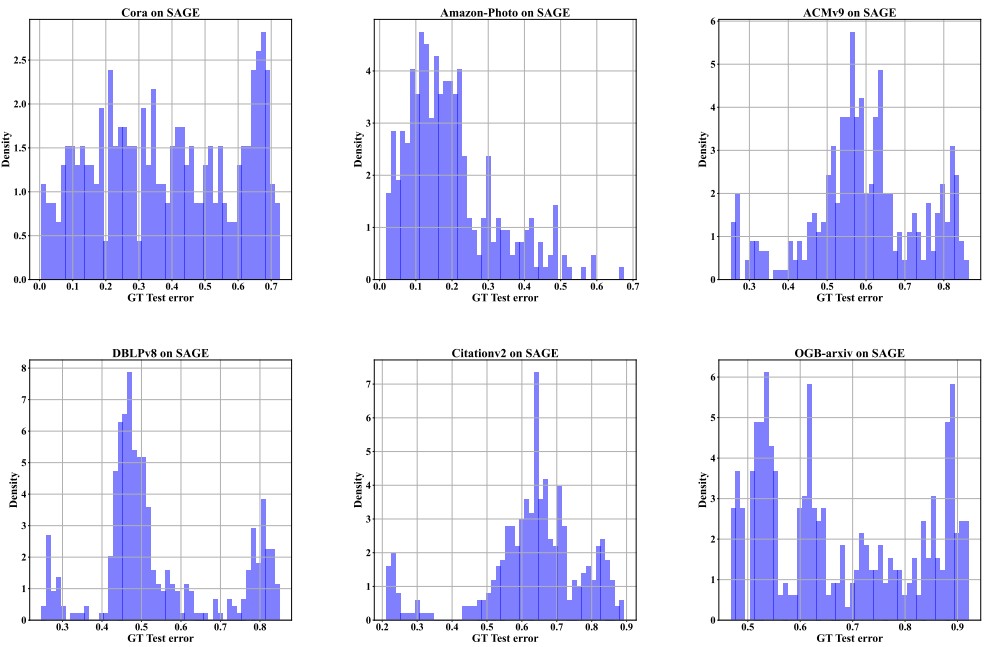

Figure A3: Ground-truth test error distributions of all test-time graphs on well-trained SAGEs.

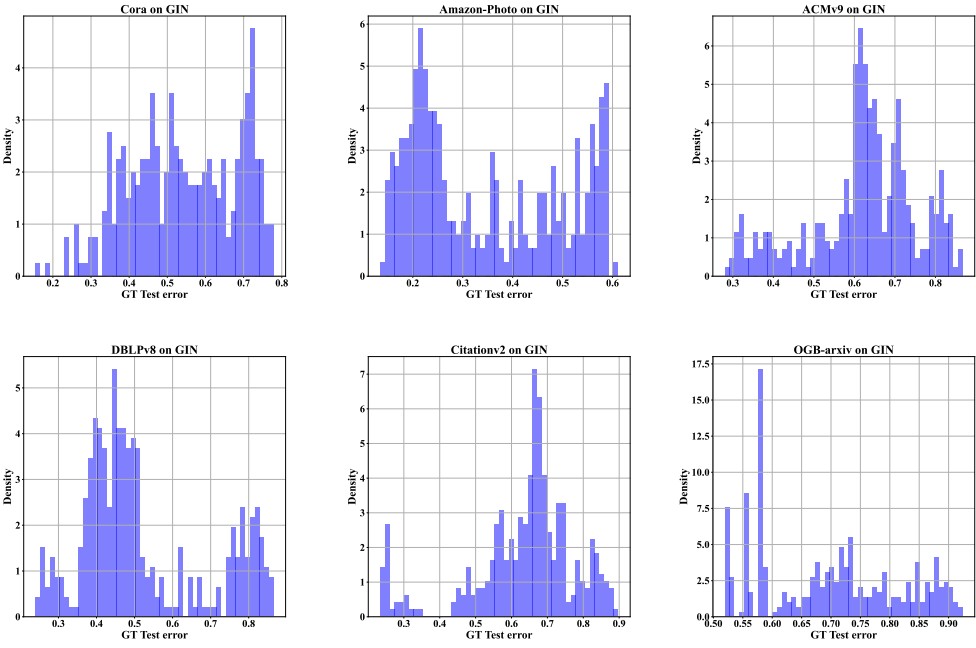

Figure A4: Ground-truth test error distributions of all test-time graphs on well-trained GINs.

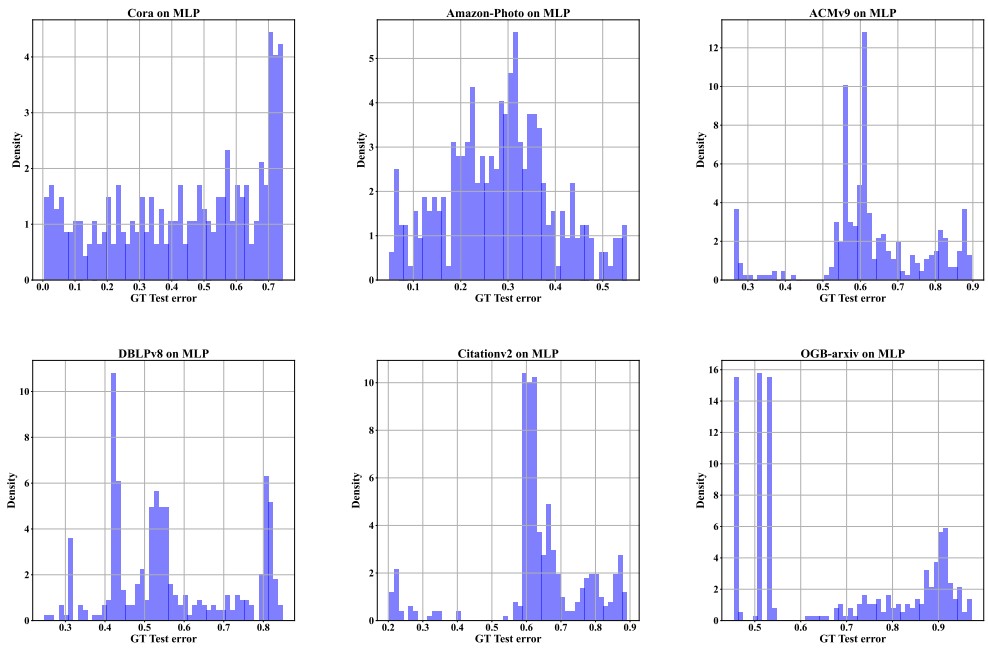

Figure A5: Ground-truth test error distributions of all test-time graphs on well-trained MLPs.

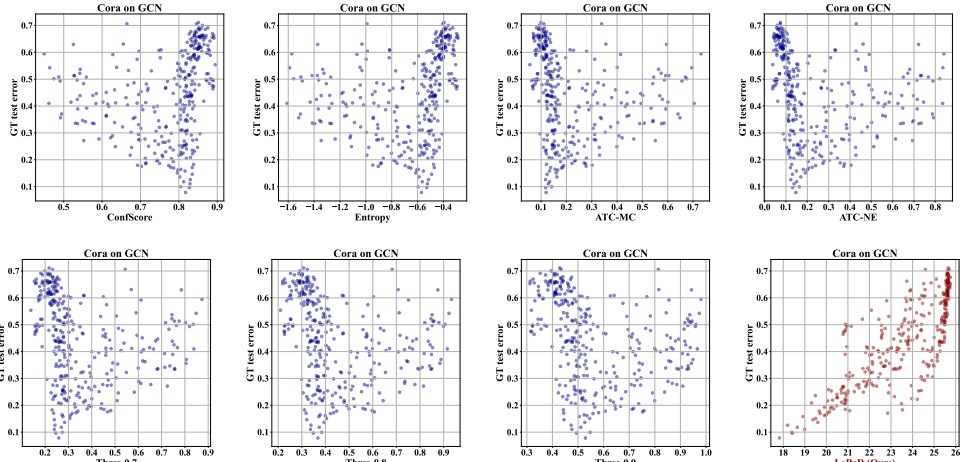

Figure A6: Correlation visualization comparisons between existing methods and our proposed LEBED on Cora with well-trained GCNs.

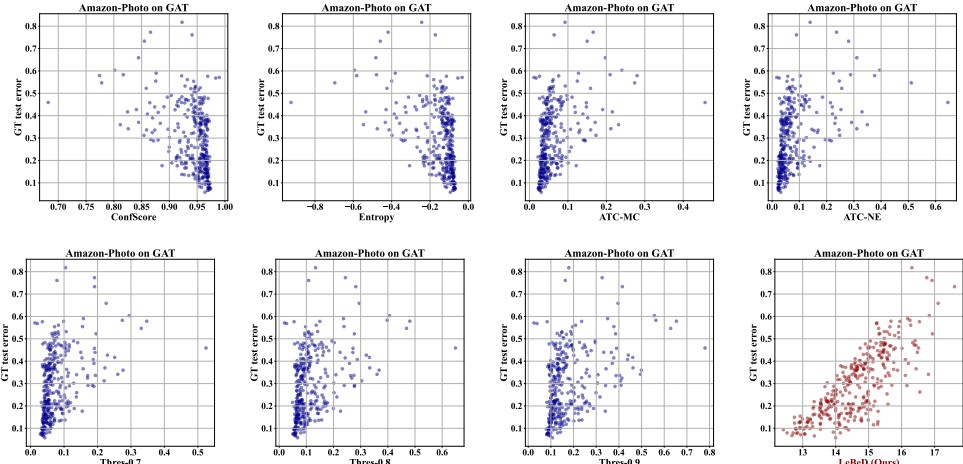

Figure A7: Correlation visualization comparisons between existing methods and our proposed LEBED on Amazon-Photo with well-trained GATs.

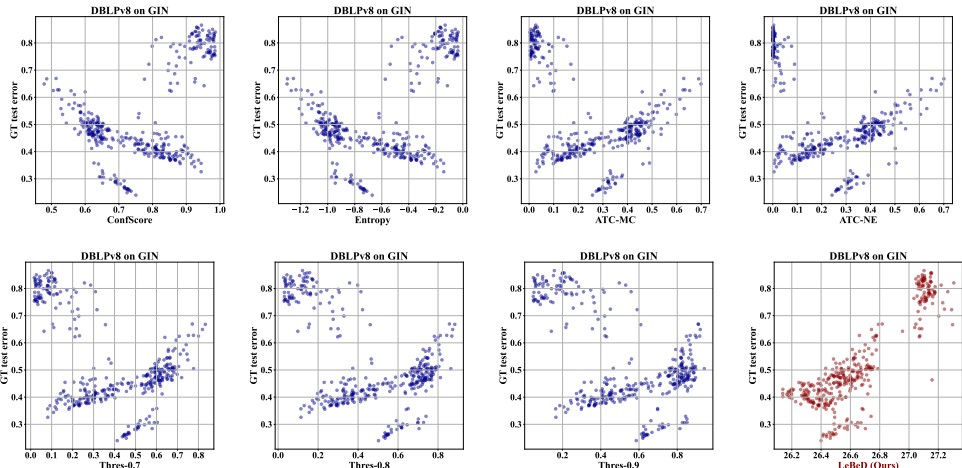

Figure A8: Correlation visualization comparisons between existing methods and our proposed LEBED on DBLPv8 with well-trained GINs.

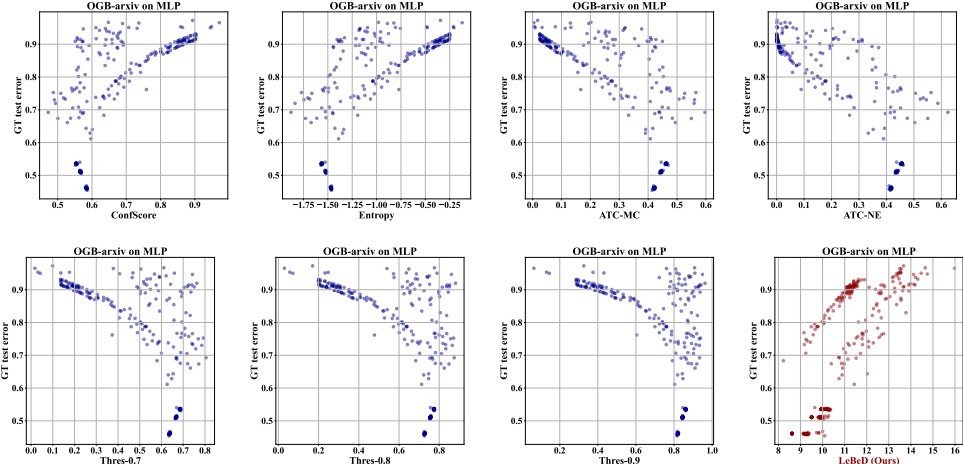

Figure A9: Correlation visualization comparisons between existing methods and our proposed LEBED on OGB-arxiv with well-trained MLPs.

