# OpenReview forum: "Online GNN Evaluation Under Test-time Graph Distribution Shifts"
_ICLR.cc/2024/Conference — ICLR 2024 spotlight_

### Official Review · Reviewer_mCrw · 2023-10-30

**Soundness:** 3 good
**Presentation:** 3 good
**Contribution:** 4 excellent
**Rating:** 8
**Confidence:** 4

**Summary:**

The paper explores the challenge of evaluating well-trained Graph Neural Network (GNN) models in real-world scenarios where test graph data lacks labels and may differ from the training data's distribution. To address this, the authors propose "Learning Behavior Discrepancy" (LEBED) as a score to assess a GNN's generalization ability under test-time distribution shifts. LEBED measures differences in learning behavior between a GNN trained on the training graph and one retrained on the test graph, considering both node prediction and structure reconstruction discrepancies. A novel GNN re-training strategy is introduced for this purpose. The paper's experiments demonstrate that the LEBED score strongly correlates with ground-truth test errors for various GNN models, offering an effective evaluation metric. The paper acknowledges certain assumptions and suggests addressing more complex distribution shifts in future research.

**Strengths:**

1.	Novel Research Problem: The paper addresses a novel research problem, online GNN evaluation. Previous works only consider test the GNN performance after training it. This paper proposes a new research problem that how to evaluate GNN’s performance under distribution shifts and no access to the original training data. From my perspective, it is an important and practical concern in the field of graph neural networks. This is a significant contribution as it explores the evaluation of GNNs under real-world conditions with distribution shifts and the absence of test node labels. It also paves the way to pre-training large-scale GNN models.

2.	Proposed Metric (LEBED): The paper introduces a novel metric, LEBED (Learning Behavior Discrepancy), which aims to estimate the generalization error of well-trained GNN models under distribution shifts. It re-trains the GNN under test data and then calculate the discrepancy of optimal weights between training and test data. Fig 2 gives a clear framework.

3.	Experimental Validation: The paper conducts extensive experiments on real-world test graphs under diverse distribution shifts, demonstrating the effectiveness of the proposed method. Datasets of various distribution shifts scenarios are provided, including node feature shifts, domain shifts and temporal shifts.

**Weaknesses:**

1.	Table 1 shows the test-time graph data. I wonder where did you get the pre-trained GNNs? What datasets were used for pre-training?
2.	In S2, GNN is re-trained was test data. However, if the same training data is used during S2, it will also cause difference of the optimal weights due to the randomness. Will this affect the LeBed score? In other words, is part of LeBed score caused by the randomness of training process?
3.	Eq. 4 shows that Psuedo label $\mathbf{Y}_{\mathrm{te}}^*$ is from pre-trained GNN with test-time data, which has distribution shifts from original training data. Why is it can be used for the supervision of re-training? More explanations are expected here.

**Questions:**

See weaknesses.

---

> ### Author Response · Authors · 2023-11-13
> **Response to Reviewer mCrw**
>
> We sincerely appreciate your thoughtful review of our paper. We are glad to hear that you recognize the significance of online GNN evaluation problem under test-time graph distribution shifts. We have carefully considered your comments and suggestions, and the following are our detailed responses. We are expecting these could help answer your questions.
> ***
> **W1-[Datasets for Pretraining GNN]**
>
> We obtained the pre-trained GNNs with the "#Training Graph" for all datasets in Table 1 of our manuscript.
>
> For instance, for the first row "Node feature shifts, Cora" with  "#Train/Val/Test Graphs= 1/1/320 (8-artifices)", we use the 1 training graph of the original Cora dataset to pre-train the GNNs and 1 its validation graph to indicate the optimality of the pre-trained GNNs. Then, during the test time, we evaluate our proposed LEBED score on the 320 test graphs with 8 different distinct cases of artificial feature shifts on Cora.
>
> Likewise, for "Domain shifts ACMv9", we also use a single training graph of original ACMv9 for pretraining GNNs with its validation graph, and during the test time, we evaluate our LEBED on 369 test graphs covering three domains of  ACMv9, DBLPv8, Citationv2.
>
> For "Temporal shifts OGB-arxiv", as mentioned in "Appendix B TEST-TIME DATASET DETAILS", the training graph for pre-trained GNNs is the year range [1950, 2011] corresponding subgraph of the original OGB-arxiv, [2011, 2014] subgraph for validation. During the test time, three-year periods [2014, 2016], [2016, 2018], [2018, 2020] with 360 test graphs are used for evaluating our LEBED.
> ***
> **W2-[Training Data Performance During S2]**
>
> We would like to address this concern from two aspects:
>
> **Theoretically**, when the same training data is used in S2, generally, the "pseudo labels" of the training graph would be very similar to the ground-truth labels, since the pre-trained model has seen and used the label information. Hence, the optimal weights would be also similar when the model starts from the same initialization point without too much randomness.
>
> **Empirically**, we test the potential randomness from 10 different random seeds with the same training data for S2 on "Cora" datasets on all GNN models. We report the average LEBED score with standard deviation in Table.Re-mCrw-1. As can be observed, the randomness of the training process has minimal impact on the LEBED score, as indicated by the very low standard deviation observed across the same training set.
>
> Table.Re-mCrw-1. The proposed LEBED score on average with standard deviation (±STD) in the S2-Retraining stage over the same training set G_tr within 10 runs.
>
> | Cora (S2)                       |   GCN   |   GAT   |  SAGE   |   GIN   |   MLP   |
> |:----------------------------:|:-------:|:-------:|:-------:|:-------:|:-------:|
> | LEBED (G_tr with STD)        | 5.134±0.00 | 3.951±0.01 | 9.034±0.00 | 9.080±0.34 | 0.000±0.00 |
>
> ***
> **W3-[Psuedo Label Supervise Re-training]**
>
> In the context of "online GNN model evaluation", the evaluation of a pre-trained GNN's capabilities on an unseen and unlabeled test graph Gte relies on the outputs from the pre-trained GNN, including the pseudo labels $Y_{te}^{∗}$ and node embeddings $Z_{te}^{∗}$, as specified in Eq.(4). They serve as key reference information from the pre-trained GNN during the testing phase.
>
> During the re-training stage, the meaning of "supervision" for online GNN model evaluation is a little different from model training. Here, the pseudo labels $Y_{te}^{∗}$ are not specifically for supervising GNN to achieve the optimal performance on the test graph, but as a reference guidance to quantify the divergence between the pseudo-labels $Y_{te}^{∗}$ generated by the pre-trained GNN models vs. the predicted labels $\hat{Y}_{te}$ by the re-trained GNN models.
>
> In this case, it can be used for "the supervision of re-training", allowing for evaluating how well the re-trained models can adapt to and predict for the test graph, compared to the initial pre-trained models.

---

> > ### Comment · Reviewer_mCrw · 2023-11-14
> >
> > Thanks for your reply. The explanations and experiment results addressed my concerns well. Overall, I'd like to recommend an accept for this paper.

---

### Official Review · Reviewer_sJ17 · 2023-10-31

**Soundness:** 3 good
**Presentation:** 3 good
**Contribution:** 4 excellent
**Rating:** 8
**Confidence:** 4

**Summary:**

This work proposed a metric named LEBED, learning behavior discrepancy score, for a new research problem "online GNN evaluation" in terms of the graph distribution shift issue during test time. By imitating the learning behavior in GNN training parameters, this metric could indicate the GNN model performance without the test graph labels through the parameter space distance between the training and test graphs. Moreover, this work contains a parameter-free optimality criterion which introduces graph structure information to guide GNN retraining on the test-time graphs. Experiments under different training-test graph shifts could verify its effectiveness.

**Strengths:**

The strength of the work:

(1) The overall idea and research topic of "online GNN evaluation" is interesting and novel for understanding the real-world GNN model performance, especially the graph distribution shift issue is also considered, making it more applicable.  This can be a new research direction that inspires future work on GNN model understanding and real-world deployment.

(2) The solution with the proposed LEBED metric for estimating test-time generalization error on graph data appears logical to me. With retraining GNN on the test unlabeled graphs with distribution shifts, its parameter space metric with the context of learning behavior discrepency is reasonable for GNN model evaluation, especially given its graph structure discrepency-related criterion.

(3) The experiments involve graphs with diverse distribution shift cases, domains, and scales, making this metric relatively convincing. And the performance of this method seems effective enough as they argued to indicate the test-time GNN performance.

(4) The paper is well-structured, and easy to understand. Its logic and writing look good to me.

**Weaknesses:**

Q1: How to explain that some baseline methods with some cases positive values but other cases negative values, more explanations of the experimental results should be provided here.

Q2: How easily can the proposed LEBED be integrated into existing GNN frameworks or pipelines, considering it seems a general GNN model evaluation method covering different models and test graph data?

Q3: Does retraining the GNN on unlabeled test graphs introduce significant computational overhead, and what about the time complexity?

**Questions:**

See above.

---

> ### Author Response · Authors · 2023-11-13
> **Response to Reviewer sJ17**
>
> We sincerely appreciate your thoughtful review of our paper. We are so encouraged by your recognition of the “new research direction” for online GNN evaluation, and this means a lot to advancing the GNN deployment in real-world applications, especially under the graph distribution shift scenarios. We have carefully considered your comments and suggestions, and the following are our detailed responses. We are expecting these could be helpful in answering your questions.
>
> ***
> **Q1-[Numeric Values of Baseline Results]**
>
> As our proposed LEBED serves as an online GNN evaluation metric, we report all the results based on the correlation between our LEBED scores and the ground-truth test error, using two correlation metrics: the linear correlation R2 (ranging from 0 to 1) and Spearman’s rank correlation (ranging from -1 to 1). Negative values in Spearman’s rank correlation indicate a negative correlation with ground-truth test errors.
>
> For example, as shown in Table 2 of our submission, the baseline method "ConfScore" has a negative correlation of -0.55 with the Amazon-Photo dataset when evaluating the GCN model, but shows a positive correlation of 0.94 when evaluating the GIN model.
>
> These results just reflect the instability and inconsistency of current baseline methods and suggest they may not be ideal or optimal for online GNN evaluation under test-time graph distribution shifts. In contrast, our proposed LEBED achieves consistently positive correlations on all test-time online evaluations over all distribution shifts and well-trained GNN models, demonstrating its effectiveness.
>
> ***
> **Q2-[Ability of Integration into Existing GNN Frameworks]**
>
> The proposed LEBED metric is designed to be a versatile measurement for evaluating online GNN models, adaptable to various GNN architectures even under various test-time graph distribution shifts. Besides, our proposed test-time GNN retraining strategy is flexible, requiring only that the GNN architecture of the training and test stages be identical, regardless of the specific architecture used. This flexibility makes it straightforward to integrate LEBED into any existing GNN model framework.
> ***
> **Q3-[Computation Costs]**
>
> In the "C.1 TIME COMPLEXITY ANALYSIS" section of Appendix in our submission, we outline the computational costs of retraining. These costs depend on various factors: $E$ (the number of edges), $L$ (the number of GNN layers), $q$ (the iteration number of retraining), $d$ (the input feature dimension), $M$ (the number of test graph nodes), and $w$ (the number of overall GNN parameters). The retraining computation costs can be represented as $O(\Omega) + O(q \cdot \Omega) + O(M^2) + O(w)$, where $O(\Omega) = O(LMd^2) + O(E).$
> This process is primarily influenced by the size of the test graphs and the retraining iterations. Generally, the computation costs during test time are not excessive but are comparable to the original GNN training process on the training graph.

---

### Official Review · Reviewer_CD29 · 2023-11-01

**Soundness:** 3 good
**Presentation:** 3 good
**Contribution:** 3 good
**Rating:** 8
**Confidence:** 4

**Summary:**

This paper studies the training-test graph data distribution shift problem along with GNN model evaluation problem.
Its method contains a GNN retraining part and a parameter-free optimality criterion. It develops a metric LEBED to estimate the test-time errors of trained GNN models, and this proposed metric can be taken as an indicator to show the performance of GNN on real-world test graphs without labels. Its experimental parts involves many evaluation models and graph datasets with distribution shifts covering node feature, domain, and temporal, the experimental results could demonstrate the metric's effectiveness under the online test-time GNN evaluation scenario.

**Strengths:**

1. The proposed online GNN evaluation problem under test-time graph distribution shift is insightful. It attempts to understand the well-trained GNN models' performance in the practical GNN model deployment and service scenario, which could be useful to real-world GNN development.

2. The introduction of the proposed evaluation metric is thoughtful with a relatively clear problem definition. The main idea of leveraging the GNN retraining discrepency to align the training-test graph discrepancy is comprehensible when the online scenario does not allow to access the training graphs.

3. The paper tests and evaluates their proposed LEBED metric on a wide range of real-world potential graph distribution shift datasets, it could provide empirical evidence of its practical utility as they claimed. Their findings can be interesting in that some existing CNN evaluation metrics would not commit consistent performance for GNNs.

**Weaknesses:**

1. In the dataset part, could you provide more details about how the graph distribution shifts are simulated in the experiments? And can these test graph datasets be publicly released?

2. Given the proposed GNN re-training process on test-time graphs, is the parameter-free criterion in Eq.(6) used for stopping the retraining but not being optimized in retraining? More explanations are needed here.

3. How is the ground-truth performance (test error) of these distribution-shifted test graphs on the online GNN models?

**Questions:**

see weakness

---

> ### Author Response · Authors · 2023-11-13
> **Response to Reviewer CD29**
>
> Thanks for your insightful and constructive review of our work. We especially appreciate your interest in online test-time distribution shift scenarios and our proposed online GNN evaluation problem. We are encouraged to know that our efforts on "GNN retraining discrepency to align the training-test graph discrepancy" have been recognized. Following are our responses, and we are expecting these could help answer your questions.
> ***
>
> **W1-[Graph Distribution Shift Simulation Details]**
>
> We have listed how we simulate the graph distribution shifts in "Appendix B TEST-TIME DATASET DETAILS" in the submission. Generally, we follow the same original dataset train/validation/test splits as Wu et al. (2022) for "node feature shifts" and "temporal shifts", and the inductive version of  Wu et al. (2020) for "domain shifts".
>
> We expand all raw test graphs with four types of strategies, including feature perturbation with Gaussian covariate shifts, feature masking, sub-graph sampling, and edge drop, with the default random range [0.1, 0.7] in uniform distribution sampling with different quantities. More detailed statistical information can be found in Table A1 of the Appendix in our submission.
>
> We are happy to make these distribution-shifted test graphs publicly available for advancing the development of the new topic of "online GNN evaluation" after the review.
> ***
>
> **W2-[Role of Parameter-free Criterion in Retraining]**
>
> The parameter-free criterion in Eq.(6) is used for the stop criterion and does NOT participate in the retraining optimization during the test time. We would like to provide more explanations as follows (also Ref. Page-5 paragraph behind Eq.(7) in the manuscript).
>
> The principle of our proposed LEBED score is to measure the learning behavior discrepancy in GNN model's parameter space. To ensure an accurate comparison of optimal parameters between $GNN_{tr}$ (the training GNN) and $GNN_{te}$ (the test-time GNN), it's essential to maintain consistent learning objectives.
>
> Typically, the training of $GNN_{tr}$, which may not be available during online test time evaluation, is focused solely on a classification objective. This objective involves $D_{Pred.}$, represented by cross-entropy loss, and does not typically encompass a self-supervised objective like $D_{Stru.}$ Therefore, to maintain alignment with the original training objectives of $GNN_{tr}$, we opt to update the retrained GNN during test time using only $D_{Pred.}$ as the objective function.
>
> ***
> **W3-[Ground-truth Test Graph Performance Under Distribution Shifts]**
>
> We have included the ground-truth test error distribution histograms of all test graphs over all well-trained GNN models in Fig. A1 to Fig. A5 in the Appendix of our submission.
>
> These figures demonstrate the extensive range of test error scenarios our evaluation protocol can encompass, when applied to online test graphs, particularly under various distribution shifts across different GNN models.
>
> For example, Fig. A1 showcases that the online Cora test graphs with node feature shifts, exhibit a broad range of ground-truth test errors, from 0.1 to 0.7, on the well-trained GNN models. These wide variations in test errors reflect that our experimental results could cover comprehensive online GNN evaluation cases under test-time distribution shifts. We have added the corresponding analysis to Appendix D3 in the final version.

---

### Official Review · Reviewer_83YC · 2023-11-01

**Soundness:** 3 good
**Presentation:** 4 excellent
**Contribution:** 3 good
**Rating:** 8
**Confidence:** 4

**Summary:**

In this work, the authors proposed a new online GNN evaluation problem, which aims to provide the reference for practical GNN applications on real-world test graphs under test-time distribution shifts. To achieve the goal of estimating the test-time errors, the author proposed a novel GNN retraining strategy with a parameter-free optimality criterion, and this could lead to an effective LEBED score, as the output and the metric to evaluate well-trained GNNs in online setting. Experiments cover diverse distribution shifts, and on different gnn models, it also show correlation with true test error, demonstrating its effectiveness.

**Strengths:**

S1: Problem: online GNN evaluation is a relatively fresh research topic to me, I think the problem that the author identified is novel. And the evaluation definition, setting, and graph distribution shift types, are clear.

S2: Methodology: the author retrained previously trained GNN on the test graph and used a criterion to indicate the GNN is well optimized in a parameter-free manner. Following this, the distance between GNN training parameters is utilized to calculate the LEBED score. I think the entire methodology looks clear and with soundness.

S3: Experiments: the experiments are well designed, and the author attempted to cover different test graph distribution shift cases, which is good and looks convincing. The results in terms of Spearman rank correlation rou and linear fitting R2 well presents the effectiveness of LEBED. Different GNN models are covered in their evaluation. Ablation study, running time, hyperparameter sensitivity, correlation visualization are properly provided.

**Weaknesses:**

W1: in page 4, the authors argued that DPred. is used as the supervision signal for instructing the GNN retraining and DStru. as the self-supervison stop criterion, and according to my understanding, it might be not hard to use both of them to instruct GNN retraining, why only use the Dpred?

W2: in page 5 stage2, it seems like the proposed method requires the retraining process have the same initialization GNN model parameters? how to make sure this point?

W3: for the domain shift experiments, the author used 1/1/369 splits with three domains A, D, C in table.1, for example, does this mean, the training and validation implements on ACM dataset, but 369 test graphs from D and C? this part is not clearly clarfied enough.

**Questions:**

See weaknesses.

---

> ### Author Response · Authors · 2023-11-13
> **Response to Reviewer 83YC**
>
> We sincerely appreciate your valuable suggestions and comments on our work, and we are pleased to learn that the practical value of our proposed LEBED score for online GNN evaluation is positively identified by the reviewer. The following are our detailed responses to the reviewer’s thoughtful comments. We are expecting these could be helpful in answering your questions.
>
> ***
> **W1-[$D_{Pred.}$ for GNN retraining]**
>
> We agree that it is possible and feasible to incorporate both $D_{Pred.}$ and $D_{Stru.}$ into the test-time GNN retraining optimization objective, but we still choose to only use $D_{Pred.}$ as learning objective and $D_{Stru.}$ as the stop criterion for the following reasons (Ref. Page-5 paragraph behind Eq.(7) in the manuscript):
>
> The principle of our proposed LEBED score is to measure the learning behavior discrepancy in GNN model's parameter space.
> To ensure an accurate comparison of optimal parameters between $GNN_{tr}$ (the training GNN) and $GNN_{te}$ (the test-time GNN), it's essential to maintain consistent learning objectives.
>
> Typically, the training of $GNN_{tr}$, which may not be available during online test time evaluation, is focused solely on a classification objective. This objective involves $D_{Pred.}$, represented by cross-entropy loss, and does not typically encompass a self-supervised objective like $D_{Stru.}$ Therefore, to maintain alignment with the original training objectives of $GNN_{tr}$, we opt to update the retrained GNN during test time using only $D_{Pred.}$ as the objective function.
>
> ***
> **W2-[Same GNN Initialization]**
>
> Yes, our proposed LEBED score necessitates that the $G_{te}$ retraining process during test time begin with the same GNN parameter initialization as used in the original $G_{tr}$ training process.
>
> To achieve this, it's important to preserve the initialization parameters of the well-trained GNN models. These parameters should then be reapplied in the online test time evaluation to ensure the retraining process to be optimized from the same start point as that in the training stage.
>
> ***
>
> **W3-[Domain Shift Dataset Details]**
>
> For "Domain shifts ACMv9", we use a single training graph of original ACMv9 for pretraining GNNs, while a single validation graph of original ACMv9 for indicating its optimality.
>
> During the test phase, we evaluated our LEBED score across 369 test graphs. These graphs spanned three distinct domains: ACMv9, DBLPv8, and Citationv2, thereby encompassing a wide range of domain shifts. And as provided details in Table A1 of the Appendix, we have test graphs with ACMv9=#41, DBLPv8=#164, and Citationv2=#164 (total: 369).

---

### Author Response · Authors · 2023-11-13
**Common Response to All Reviewers**

We thank all reviewers for their thorough review and valuable suggestions. We are delighted that our contributions have been positively acknowledged, including:

**(1) Novel research question for new exploration of online GNN model evaluation problem, along with its practical applicability for real-world GNN deployment ( @All Reviewers!)**

**(2) Novel, clear, and thoughtful Learning Behavior Discrepancy, LEBED score, as metric with logical retraining strategy and parameter-free optimality criterion,  for addressing online GNN evaluation under test-time graph distribution shifts (@All Reviewers!)**

**(3) The experimental setting covers various test graph distribution shifts scenarios, aligning with real-world applications(@All Reviewers!), convinced and effective performance (Reviewer sJ17, Reviewer mCrw), comprehensive analysis (Reviewer 83YC, Reviewer CD29).**

**(4) Clear framework (Reviewer mCrw), comprehensible design for online scenario (Reviewer CD29), well-structured paper (Reviewer sJ17).**

We greatly appreciate all the positive comments on our work. These comments encourage us to continue our efforts in advancing this very young research area of online GNN evaluation under test-time graph distribution shifts. We are expecting this work can be beneficial to advance the GNN model inference and deployment in real-world applications.

More detailed responses are as follows. We hope our responses address all weaknesses and questions! Please let us know if there is any concern. We have considered your valuable suggestions, and have modified accordingly to improve the manuscript in the final version.

---

### Meta-Review · Area_Chair_DQsM · 2023-12-06

**Metareview:**

This paper considers the problem of evaluating online GNNs, specifically tackling the issue of train-test distribution shift. The contributions of the paper revolve around solving this problem specifically by taking into account real-world constraints, such as inability to access the train graph at test time. The solution given is dubbed the "LEBED" metric for estimating the test-time generalization error.

The reviewers unininmously found this to be a good paper. The stated weaknesses were mostly requests for clarifications and the authors have provided an adequate rebutal.

Key strengths:
* Offering insights about train/test time distribution shift, including characterizing in the first place the issues with online GNN evaluation in real-world scenarios
* LEBED is a method that the reviewers find reasonable and adequately desscribed
* Experiments are convincing

Key weaknesses:
* As part of the solution, there is a required GNN retraining step.
* The test scenario considered is quite specific (e.g. no access to training graph), but at the same time is realistic in certain applications, e.g. online setting, which is what this paper focuses on.

**Justification For Why Not Higher Score:**

In many real-world applications GNNs are not deployed necessarily in an online manner and/or it might not be feasible to perform re-training steps, instead inductive inference might be used for the new nodes. This is not to say that the paper does not have valuable contributions, but the community that might find it interesting is not as broad.

**Justification For Why Not Lower Score:**

It is a well-written paper, the setting where it is applied is made clear by the authors (online GNNs, well-trained GNN etc) and the knowledge added by this work is significant. The reviewers are unanimously in favour.

---

### Decision · Program_Chairs · 2024-01-16

Accept (spotlight)